# Conditional Score Guidance for Text-Driven Image-to-Image Translation

**Hyunsoo Lee**[1*]    **Minsoo Kang**[1*]    **Bohyung Han**[1,2]

[1]ECE & [2]IPAI, Seoul National University

{philip21, kminsoo, bhhan}@snu.ac.kr

## Abstract

We present a novel algorithm for text-driven image-to-image translation based on a pretrained text-to-image diffusion model. Our method aims to generate a target image by selectively editing regions of interest in a source image, defined by a modifying text, while preserving the remaining parts. In contrast to existing techniques that solely rely on a target prompt, we introduce a new score function that additionally considers both the source image and the source text prompt, tailored to address specific translation tasks. To this end, we derive the conditional score function in a principled way, decomposing it into the standard score and a guiding term for target image generation. For the gradient computation about the guiding term, we assume a Gaussian distribution for the posterior distribution and estimate its mean and variance to adjust the gradient without additional training. In addition, to improve the quality of the conditional score guidance, we incorporate a simple yet effective mixup technique, which combines two cross-attention maps derived from the source and target latents. This strategy is effective for promoting a desirable fusion of the invariant parts in the source image and the edited regions aligned with the target prompt, leading to high-fidelity target image generation. Through comprehensive experiments, we demonstrate that our approach achieves outstanding image-to-image translation performance on various tasks. Code is available at https://github.com/Hleephilip/CSG.

## 1 Introduction

Diffusion models [1–4] have recently shown remarkable performance in various tasks such as unconditional generation of text [5, 6] or images [7, 8] and conditional generation of images [9–11], 3D scenes [12, 13], motion [14, 15], videos [16, 17], or audio [18, 19] given text and/or images. Thanks to large-scale labeled datasets of text-image pairs [20–24], text-to-image diffusion models [25–29] have been successfully trained and have achieved outstanding performance. Despite the success of the text-to-image generation models, it is not straightforward to extend the models to text-driven image-to-image translation tasks due to the limited controllability on the generated images. For example, in the case of the "cat-to-dog" task, naïve text-to-image diffusion models often fail to focus on the area of cats in a source image for updates and simply generate a dog image with a completely different appearance.

To sidestep such a critical problem, existing image-to-image translation methods [30–33, 11, 34, 35] based on diffusion models aim to update the semantic content in the source image specified by a given condition while preserving the region irrelevant to the condition for target image generation. For example, [31, 34] fine-tune the pretrained text-to-image diffusion model to reduce the distance between the background in the source image and the generated target image, while bringing the translated image closer to the target domain. On the other hand, [30, 32, 33, 11, 35] revise the

---

*Both authors contributed equally.

37th Conference on Neural Information Processing Systems (NeurIPS 2023).

generative processes based on the pretrained diffusion models for the text-driven image editing tasks without extra training.

The main idea of this work is to estimate a score function conditioned on both the source image and source text in addition to the standard condition with the target text. The new score function is composed of two terms: (a) the standard score function conditioned only on the target prompt and (b) the guiding term, *i.e.*, the gradient of the log-posterior of the source latent given the target latent and the target prompt with respect to the target latent. Note that the posterior is modeled by a Gaussian distribution whose mean and covariance matrix are estimated without an additional training process. We also employ an effective mixup strategy in cross-attention layers to achieve high-fidelity image-to-image translation. The main contributions of our paper are summarized as follows:

- We mathematically derive a conditional score that provides guidance for controllable image generation in image-to-image translation tasks; the score can be conveniently incorporated into existing text-to-image diffusion models.

- We propose a novel cross-attention mixup technique based on text-to-image diffusion models for image-to-image translation tasks, which adaptively combines the two outputs of cross-attention layers estimated with the latent source and target images at each time step.

- We introduce an intuitive performance evaluation metric that measures the fidelity of pairwise relationships between images before and after translation. Experimental results verify that our method consistently achieves outstanding performance on various tasks with respect to the standard and the proposed metrics.

The rest of the paper is organized as follows. Section 2 reviews the related work and Section 3 discusses basic concepts and practices of image-to-image translation based on diffusion models. The details of our approach are described in Section 4, and the experimental results are presented in Section 5. Finally, we conclude this paper in Section 6.

## 2 Related Work

This section describes existing text-to-image diffusion models and their applications to image-to-image translation tasks.

### 2.1 Text-to-Image Diffusion Models

Diffusion models [1–4] have achieved exceptionally promising results in text-to-image generation. For example, Stable Diffusion and Latent Diffusion Models (LDM) [26] employ a two-stage framework [36, 37], where an input image is first projected onto a low-dimensional feature space using a pretrained encoder and a generative model is trained to synthesize the feature conditioned on a text embedding. At inference time, the image is sampled by generating the representation and then decoding it. In the two-stage framework, Denoising Diffusion Probabilistic Model (DDPM) [2] is employed as the generative model while a text prompt is represented using a text encoder given by Contrastive Language-Image Pretraining (CLIP) [38]. On the other hand, DALL-E 2 [28] consists of two main components: a prior model and a decoder. The prior model infers a CLIP image embedding from a text representation encoded by the CLIP text encoder while the decoder synthesizes an image conditioned on both its CLIP embedding and text information. In addition, Imagen [25] generates an image conditioned on a text embedding provided by a powerful language encoder [39], which is trained on text-only corpora to faithfully represent a text prompt.

### 2.2 Text-Driven Image-to-Image Translation Methods using Diffusion Models

Existing image-to-image translation methods [30–32, 34, 33, 11, 35] aim to preserve the background while editing the object part only. Specifically, Stochastic Differential Editing (SDEdit) [30] infers a latent variable at an intermediate time step based on a source image and then synthesizes a target image by solving the reverse time stochastic differential equation from the intermediate time step to the origin. DiffusionCLIP [31] fine-tunes a text-to-image generative network using the local directional loss [40] computed from CLIP [38] while this method also employs the $L_1$ reconstruction loss for preserving the background and optionally the face identity loss [41] only for human face image manipulation tasks. Also, Imagic [34] optimizes a pretrained text-to-image diffusion model

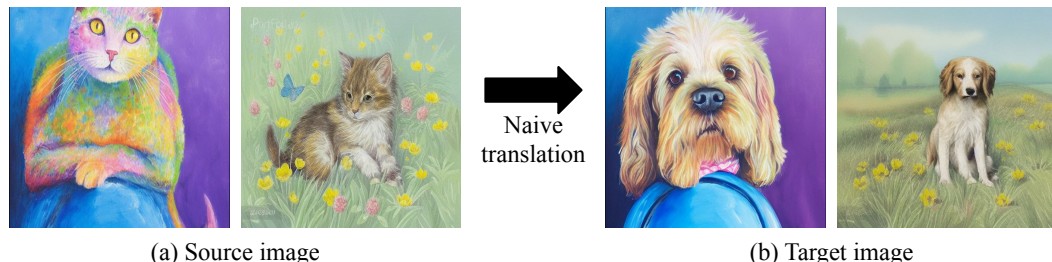

(a) Source image                                 (b) Target image

Figure 1: Image translation results of $\mathbf{x}_0^{\mathrm{tgt}}$ based on $\mathbf{x}_0^{\mathrm{src}}$ using the simple naïve DDIM generative process based on Eq (3).

conditioned on the inferred text feature to faithfully reconstruct a given image. Using the fine-tuned model, at inference time, the target image is synthesized based on the linear combination of the two features of the predicted source and the target text. In contrast, Blended Diffusion [32] requires a user-provided background mask so that the latents of the source and target images are mixed according to the mask to preserve the background while selectively updating the region of interest.

## 3 Preliminary: Image-to-Image Translation based on Diffusion Models

### 3.1 Estimation of Latent Variables for Source Images

Diffusion models [1, 2, 4] estimate a series of latent variables $\mathbf{x}_1, \mathbf{x}_2, \cdots \mathbf{x}_T$ with the same dimensionality as a data sample $\mathbf{x}_0 \in \mathbb{R}^{H \times W \times C}$, where the forward sampling process of $\mathbf{x}_t$ depends on $\mathbf{x}_{t-1}$ and $\mathbf{x}_0$ in DDIM [4] while DDPM [2] assumes the data generating process follows a Markov chain. For image-to-image translation, existing works [33, 11, 35] often employ a deterministic inference using DDIM instead of a stochastic method such as DDPM, which is given by

$$\mathbf{x}_{t+1}^{\mathrm{src}} = \sqrt{\alpha_{t+1}} f_\theta(\mathbf{x}_t^{\mathrm{src}}, t, \mathbf{y}^{\mathrm{src}}) + \sqrt{1 - \alpha_{t+1}} \epsilon_\theta(\mathbf{x}_t^{\mathrm{src}}, t, \mathbf{y}^{\mathrm{src}}), \tag{1}$$

where $\mathbf{x}_t^{\mathrm{src}}$ and $\mathbf{y}^{\mathrm{src}}$ denote the latent variable of a source image $\mathbf{x}_0^{\mathrm{src}}$ and the text embedding of a source prompt using pretrained models [38, 39], and $\alpha_t \in (0, 1]$ is an element in a predefined decreasing sequence. In the above equation, $\epsilon_\theta(\cdot, \cdot, \cdot)$ is a noise prediction network parametrized with a U-Net backbone [42] and $f_\theta(\cdot, \cdot, \cdot)$ is defined as

$$f_\theta(\mathbf{x}_t, t, \mathbf{y}) := \frac{\mathbf{x}_t - \sqrt{1 - \alpha_t} \epsilon_\theta(\mathbf{x}_t, t, \mathbf{y})}{\sqrt{\alpha_t}}. \tag{2}$$

The last latent variable $\mathbf{x}_T^{\mathrm{src}}$ is sampled recursively using Eq. (1), from which the reverse process starts for the generation of the target image or the reconstruction of the source image. Note that, for simplicity, we write equations for each channel of $\mathbf{x}_t$ because the operations in all channels are identical.

### 3.2 Generative Process of Target Images

A naïve image-to-image translation technique based on diffusion models simply generates the target image $\mathbf{x}_0^{\mathrm{tgt}}$ from $\mathbf{x}_T^{\mathrm{tgt}}$, which is set to $\mathbf{x}_T^{\mathrm{src}}$, recursively using the reverse DDIM process as follows:

$$\mathbf{x}_{t-1}^{\mathrm{tgt}} = \sqrt{\alpha_{t-1}} f_\theta(\mathbf{x}_t^{\mathrm{tgt}}, t, \mathbf{y}^{\mathrm{tgt}}) + \sqrt{1 - \alpha_{t-1}} \epsilon_\theta(\mathbf{x}_t^{\mathrm{tgt}}, t, \mathbf{y}^{\mathrm{tgt}}), \tag{3}$$

where $\mathbf{y}^{\mathrm{tgt}}$ denotes the text embedding of the target prompt. Although the naïve DDIM-based translation guarantees the cycle-consistency as discussed in [43], the simple translation algorithm often results in poor generation results, failing to maintain the content structure and the background information that should be invariant to translation, as presented in Figure 1.

To address this issue, previous approaches employ the information obtained from the reverse process of $\mathbf{x}_t^{\mathrm{src}}$ when synthesizing the target image. Specifically, to preserve the background in the source image during the early steps in the reverse process of $\mathbf{x}_t^{\mathrm{tgt}}$, Prompt-to-Prompt [33] replaces the cross-attention maps of the target latents with those obtained from the source latents while Plug-and-Play [11] injects the self-attention and intermediate feature maps into the matching layers in

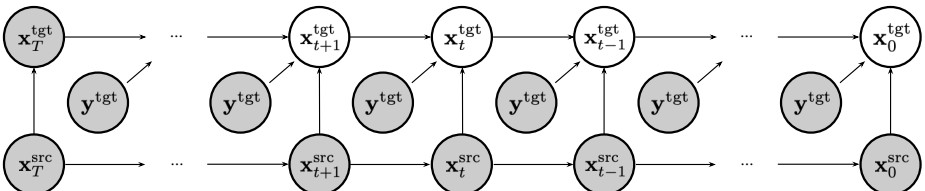

Figure 2: Graphical model of the proposed method for target image generation.

the noise prediction network. For the remaining steps, the reverse DDIM process defined as Eq. (3) is simply employed to generate the target image without any modification. On the other hand, Pix2Pix-Zero [35] first updates the latent $\mathbf{x}_t^{\text{tgt}}$ to minimize the distance between the cross-attention maps given by the reverse process of $\mathbf{x}_t^{\text{src}}$ and $\mathbf{x}_t^{\text{tgt}}$, and then performs the DDIM generation process using the updated latent.

## 4 Conditional Score Guidance with Cross-Attention Mixup

This section discusses how to derive our conditional score guidance using cross-attention mixup for high-fidelity text-driven image-to-image translation.

### 4.1 Overview

The naïve reverse process of DDIM for image-to-image translation can be rewritten by plugging Eq. (2) into Eq. (3) and adopting the approximate score function suggested in [3] as follows:

$$\mathbf{x}_{t-1}^{\text{tgt}} = \frac{\sqrt{\alpha_{t-1}}}{\sqrt{\alpha_t}}\mathbf{x}_t^{\text{tgt}} - \sqrt{1-\alpha_t}\gamma_t\epsilon_\theta(\mathbf{x}_t^{\text{tgt}}, t, \mathbf{y}^{\text{tgt}}) \tag{4}$$

$$\approx \frac{\sqrt{\alpha_{t-1}}}{\sqrt{\alpha_t}}\mathbf{x}_t^{\text{tgt}} + (1-\alpha_t)\gamma_t\nabla_{\mathbf{x}_t^{\text{tgt}}}\log p(\mathbf{x}_t^{\text{tgt}}|\mathbf{y}^{\text{tgt}}), \tag{5}$$

where

$$\nabla_{\mathbf{x}_t^{\text{tgt}}}\log p(\mathbf{x}_t^{\text{tgt}}|\mathbf{y}^{\text{tgt}}) \approx -\frac{1}{\sqrt{1-\alpha_t}}\epsilon_\theta(\mathbf{x}_t^{\text{tgt}}, t, \mathbf{y}^{\text{tgt}}) \ \text{ and } \ \gamma_t := \sqrt{\frac{\alpha_{t-1}}{\alpha_t}} - \sqrt{\frac{1-\alpha_{t-1}}{1-\alpha_t}}. \tag{6}$$

Our approach designs a new reverse process guided by the proposed conditional score, for which we replace the original score function, $\nabla_{\mathbf{x}_t^{\text{tgt}}}\log p(\mathbf{x}_t^{\text{tgt}}|\mathbf{y}^{\text{tgt}})$, in Eq. (5), with the one conditioned on the source information, $\nabla_{\mathbf{x}_t^{\text{tgt}}}\log p(\mathbf{x}_t^{\text{tgt}}|\mathbf{x}_0^{\text{src}}, \mathbf{y}^{\text{tgt}}, \mathbf{y}^{\text{src}})$. The rest of this section discusses how to derive the new conditional score function, where we incorporate a novel technique, cross-attention mixup, to estimate a mask with the foreground/background probability, and further revise the conditional score function using the mask. We refer our algorithm to Conditional Score Guidance (CSG). Figure 2 depicts the graphical model of the proposed text-driven image-to-image translation process, and Algorithm 1 shows the detailed procedure of CSG.

### 4.2 Conditional Score Estimation

For the conditional reverse process of image-to-image translation, we propose the novel conditional score, $\nabla_{\mathbf{x}_t^{\text{tgt}}}\log p(\mathbf{x}_t^{\text{tgt}}|\mathbf{x}_0^{\text{src}}, \mathbf{y}^{\text{tgt}}, \mathbf{y}^{\text{src}})$, which is given by

$$\nabla_{\mathbf{x}_t^{\text{tgt}}}\log p(\mathbf{x}_t^{\text{tgt}}|\mathbf{x}_0^{\text{src}}, \mathbf{y}^{\text{tgt}}, \mathbf{y}^{\text{src}}) = \nabla_{\mathbf{x}_t^{\text{tgt}}}\log\int p(\mathbf{x}_t^{\text{tgt}}, \mathbf{x}_t^{\text{src}}|\mathbf{x}_0^{\text{src}}, \mathbf{y}^{\text{tgt}}, \mathbf{y}^{\text{src}})\,d\mathbf{x}_t^{\text{src}}$$

$$= \nabla_{\mathbf{x}_t^{\text{tgt}}}\log\int p(\mathbf{x}_t^{\text{tgt}}|\mathbf{x}_t^{\text{src}}, \mathbf{x}_0^{\text{src}}, \mathbf{y}^{\text{tgt}}, \mathbf{y}^{\text{src}})\cdot p(\mathbf{x}_t^{\text{src}}|\mathbf{x}_0^{\text{src}}, \mathbf{y}^{\text{tgt}}, \mathbf{y}^{\text{src}})\,d\mathbf{x}_t^{\text{src}}$$

$$= \nabla_{\mathbf{x}_t^{\text{tgt}}}\log\int p(\mathbf{x}_t^{\text{tgt}}|\mathbf{x}_t^{\text{src}}, \mathbf{y}^{\text{tgt}})\cdot p(\mathbf{x}_t^{\text{src}}|\mathbf{x}_0^{\text{src}}, \mathbf{y}^{\text{src}})\,d\mathbf{x}_t^{\text{src}} \tag{7}$$

$$\approx \nabla_{\mathbf{x}_t^{\text{tgt}}}\log p(\mathbf{x}_t^{\text{tgt}}|\hat{\mathbf{x}}_t^{\text{src}}, \mathbf{y}^{\text{tgt}}). \tag{8}$$

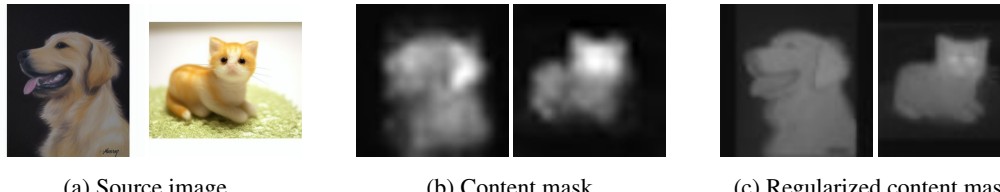

| (a) Source image | (b) Content mask | (c) Regularized content mask |

Figure 3: Visualization of the source image, the content mask $\mathbf{M}^{\mathrm{src}}[k]$, and the corresponding regularized mask $\hat{\mathbf{M}}^{\mathrm{src}}[k]$, where the $k^{\mathrm{th}}$ token in the source prompt contains the semantic information about the region to be edited in the source image.

In Eq. (7), $\mathbf{x}_t^{\mathrm{tgt}}$ is independent of $\mathbf{y}^{\mathrm{src}}$ and $\mathbf{x}_0^{\mathrm{src}}$ as illustrated in Figure 2, and $\mathbf{x}_t^{\mathrm{src}}$ is orthogonal to $\mathbf{y}^{\mathrm{tgt}}$. In Eq. (8), $\hat{\mathbf{x}}_t^{\mathrm{src}}$ is a sample drawn from $p(\mathbf{x}_t^{\mathrm{src}}|\mathbf{x}_0^{\mathrm{src}}, \mathbf{y}^{\mathrm{src}})$ using the forward deterministic inference in Eq. (1). Then, we decompose the right-hand side of Eq. (8) as

$$\nabla_{\mathbf{x}_t^{\mathrm{tgt}}} \log p(\mathbf{x}_t^{\mathrm{tgt}}|\hat{\mathbf{x}}_t^{\mathrm{src}}, \mathbf{y}^{\mathrm{tgt}}) = \nabla_{\mathbf{x}_t^{\mathrm{tgt}}} \log p(\mathbf{x}_t^{\mathrm{tgt}}|\mathbf{y}^{\mathrm{tgt}}) + \nabla_{\mathbf{x}_t^{\mathrm{tgt}}} \log p(\hat{\mathbf{x}}_t^{\mathrm{src}}|\mathbf{x}_t^{\mathrm{tgt}}, \mathbf{y}^{\mathrm{tgt}}), \qquad (9)$$

where the first term $\nabla_{\mathbf{x}_t^{\mathrm{tgt}}} \log p(\mathbf{x}_t^{\mathrm{tgt}}|\mathbf{y}^{\mathrm{tgt}})$ is estimated by the noise prediction network as the standard reverse process. Finally, the conditional score in our approach is given by

$$\nabla_{\mathbf{x}_t^{\mathrm{tgt}}} \log p(\mathbf{x}_t^{\mathrm{tgt}}|\mathbf{x}_0^{\mathrm{src}}, \mathbf{y}^{\mathrm{tgt}}, \mathbf{y}^{\mathrm{src}}) \approx \nabla_{\mathbf{x}_t^{\mathrm{tgt}}} \log p(\mathbf{x}_t^{\mathrm{tgt}}|\mathbf{y}^{\mathrm{tgt}}) + \nabla_{\mathbf{x}_t^{\mathrm{tgt}}} \log p(\hat{\mathbf{x}}_t^{\mathrm{src}}|\mathbf{x}_t^{\mathrm{tgt}}, \mathbf{y}^{\mathrm{tgt}}). \qquad (10)$$

### 4.3 Cross-Attention Mixup

To improve the derived score function, we estimate a mask $\mathbf{P}^{\mathrm{src}} \in (0, 1)^{H \times W}$ indicating where to preserve and edit in the source image, and use the mask for the computation of the score function. To this end, motivated by the observed property [33] that the spatial layout of a generated image depends on the cross-attention map, we first compute the average cross-attention maps $\mathbf{M}^{\mathrm{src}} \in \mathbb{R}^{L \times H \times W}$ of the time-dependent cross-attention maps $\{\mathbf{M}_t^{\mathrm{src}}\}_{t=1:T}$ in the noise prediction network using $\mathbf{x}_t^{\mathrm{src}}$'s as its inputs. Then, we select a content mask $\mathbf{M}^{\mathrm{src}}[k] \in \mathbb{R}^{H \times W}$, which contains the semantic information of the to-be-edited region in the source image, where $k$ denotes the position of the source prompt token to be updated in the target prompt. However, the application of the content mask $\mathbf{M}^{\mathrm{src}}[k]$ is limited to highlighting only small parts within the content of interest. For example, in the case of the "cat-to-dog" task, the head region in $\mathbf{M}^{\mathrm{src}}[k]$ exhibits high activations while the body area yields relatively low responses as illustrated in Figure 3b.

To alleviate such a drawback, our approach applies a regularization technique to the content mask $\mathbf{M}^{\mathrm{src}}[k]$ for spatial smoothness as depicted in Figure 3c. For the regularization, we compute the average self-attention map, $\mathbf{A}^{\mathrm{src}} \in \mathbb{R}^{H \times W \times H \times W}$ over time-dependent self-attention maps in the noise prediction network during the reverse DDIM process. The motivation behind this strategy is from the observation in [11] that the averaged self-attention map tends to hold semantically well-aligned attention information.

Using the content mask and the average self-attention map, we compute the regularized content mask, $\hat{\mathbf{M}}^{\mathrm{src}}[k] \in \mathbb{R}^{H \times W}$, as follows:

$$\hat{\mathbf{M}}^{\mathrm{src}}[k][h, w] := \mathrm{tr}\left(\mathbf{A}^{\mathrm{src}}[h, w](\mathbf{M}^{\mathrm{src}}[k])^T\right), \qquad (11)$$

where $\mathrm{tr}(\cdot)$ is the trace operator and $[h, w]$ denotes the pixel position. Then, the background mask $\mathbf{P}^{\mathrm{src}} \in \mathbb{R}^{H \times W}$ for image-to-image translation is given by

$$\mathbf{P}^{\mathrm{src}} := \mathbf{1} - \hat{\mathbf{M}}^{\mathrm{src}}[k], \qquad (12)$$

where $\mathbf{1} \in \mathbb{R}^{H \times W}$ denotes the matrix whose elements are 1. Each element in $\mathbf{P}^{\mathrm{src}}$ indicates the probability that the corresponding pixel in the image belongs to the region to be preserved even after translation.

Given the background mask $\mathbf{P}^{\mathrm{src}}$, we propose a new cross-attention guidance method called *cross-attention mixup* to estimate the cross-attention map for an arbitrary $\ell^{\mathrm{th}}$ text token as

$$\hat{\mathbf{M}}_t^{\mathrm{tgt}}[\ell] := \mathbf{M}_t^{\mathrm{src}}[\ell] \odot \mathbf{P}^{\mathrm{src}} + \mathbf{M}_t^{\mathrm{tgt}}[\ell] \odot (\mathbf{1} - \mathbf{P}^{\mathrm{src}}), \qquad (13)$$

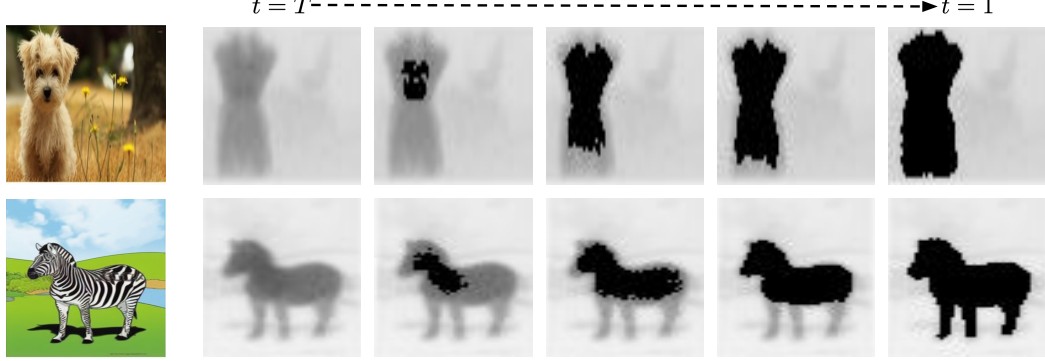

(a) Source image        (b) Precision matrix $\mathbf{\Omega}_t$ at time step $t$

Figure 4: Visualization of the source image and estimated precision matrix $\mathbf{\Omega}_t$ ranging from time step $t = T$ to $t = 1$.

---

**Algorithm 1** Conditional Score Guidance with Cross-Attention Mixup

---

**Inputs:** source image $\mathbf{x}_0^{\text{src}}$, source prompt embedding $\mathbf{y}^{\text{src}}$, target prompt embedding $\mathbf{y}^{\text{tgt}}$, hyperparameter $\lambda^{\text{pre}}$

**for** $t \leftarrow 0, \cdots, T-1$ **do**

    Compute $\mathbf{x}_{t+1}^{\text{src}}$ by Eq. (1) while saving $\mathbf{M}_t^{\text{src}}$ and $\mathbf{A}_t^{\text{src}}$

**end for**

$\mathbf{x}_T^{\text{tgt}} \leftarrow \mathbf{x}_T^{\text{src}}$

Compute $\mathbf{M}^{\text{src}}[k]$ and $\mathbf{A}^{\text{src}}$ by averaging $\mathbf{M}_t^{\text{src}}$ and $\mathbf{A}_t^{\text{src}}$ over all time steps

Compute $\hat{\mathbf{M}}^{\text{src}}[k]$ using $\mathbf{M}^{\text{src}}[k]$ and $\mathbf{A}^{\text{src}}$ by Eq. (11)

Compute $\mathbf{P}^{\text{src}} \leftarrow \mathbf{1} - \hat{\mathbf{M}}^{\text{src}}[k]$ by Eq. (12)

**for** $t \leftarrow T, \cdots, 1$ **do**

    Compute $\hat{\epsilon}_\theta(\mathbf{x}_t^{\text{tgt}}, t, \mathbf{y}^{\text{tgt}})$ using cross-attention mixup in Eq. (13)

    Compute $\gamma_t \leftarrow \sqrt{\frac{\alpha_{t-1}}{\alpha_t}} - \sqrt{\frac{1-\alpha_{t-1}}{1-\alpha_t}}$ and obtain $\mathbf{\Omega}_t$ using Eq. (15)

    Perform the proposed conditional score guidance given by Eq. (18):

    $\hat{\mathbf{x}}_t^{\text{src}} \leftarrow \mathbf{x}_t^{\text{src}}$

    $\mathbf{x}_{t-1}^{\text{tgt}} \leftarrow \sqrt{\alpha_{t-1}} \left( \frac{\mathbf{x}_t^{\text{tgt}} - \sqrt{1-\alpha_t}\hat{\epsilon}_\theta(\mathbf{x}_t^{\text{tgt}}, t, \mathbf{y}^{\text{tgt}})}{\sqrt{\alpha_t}} \right) + \sqrt{1-\alpha_{t-1}}\hat{\epsilon}_\theta(\mathbf{x}_t^{\text{tgt}}, t, \mathbf{y}^{\text{tgt}}) - \gamma_t\mathbf{\Omega}_t(\mathbf{x}_t^{\text{tgt}} - \hat{\mathbf{x}}_t^{\text{src}})$

**end for**

**Output:** target image $\mathbf{x}_0^{\text{tgt}}$

---

where $\odot$ is the Hadamard product operator and $\mathbf{M}_t^{\text{tgt}}$ denotes the cross-attention map in the noise prediction network given an input $\mathbf{x}_t^{\text{tgt}}$. Note that the cross-attention mixup is helpful to preserve the background conditioned on the text prompts while allowing us to edit the regions of interest. By integrating the cross-attention mixup in Eq. (13), we obtain the modified prediction $\hat{\epsilon}_\theta(\mathbf{x}_t^{\text{tgt}}, t, \mathbf{y}^{\text{tgt}})$ using $\hat{\mathbf{M}}_t^{\text{tgt}}$ instead of $\mathbf{M}_t^{\text{tgt}}$ at every time step $t$.

### 4.4 Conditional Score Guidance

We still need to formulate the guidance term, $\nabla_{\mathbf{x}_t^{\text{tgt}}} \log p(\hat{\mathbf{x}}_t^{\text{src}}|\mathbf{x}_t^{\text{tgt}}, \mathbf{y}^{\text{tgt}})$. To this end, we assume a Gaussian distribution for $p(\hat{\mathbf{x}}_t^{\text{src}}|\mathbf{x}_t^{\text{tgt}}, \mathbf{y}^{\text{tgt}})$ with a diagonal precision matrix $\mathbf{\Omega}_t \in \mathbb{R}^{HW \times HW}$, *i.e.*, $p(\hat{\mathbf{x}}_t^{\text{src}}|\mathbf{x}_t^{\text{tgt}}, \mathbf{y}^{\text{tgt}}) \sim \mathcal{N}(\mathbf{x}_t^{\text{tgt}}, (1-\alpha_t)\mathbf{\Omega}_t^{-1})$. The guidance term is then given by

$$\nabla_{\mathbf{x}_t^{\text{tgt}}} \log p(\hat{\mathbf{x}}_t^{\text{src}}|\mathbf{x}_t^{\text{tgt}}, \mathbf{y}^{\text{tgt}}) = -\frac{\mathbf{\Omega}_t(\mathbf{x}_t^{\text{tgt}} - \hat{\mathbf{x}}_t^{\text{src}})}{1-\alpha_t}, \tag{14}$$

where each diagonal element in $\mathbf{\Omega}_t$ has a large precision value if it corresponds to the pixel that should be preserved and consequently has a low variance, and vice versa.

Given the background mask $\mathbf{P}^{\text{src}}$ and a hyperparameter $\lambda^{\text{pre}}$ to control the magnitude of each element in $\boldsymbol{\Omega}_t$, the diagonal entries of $\boldsymbol{\Omega}_t$ are given by

$$\text{Diag}(\boldsymbol{\Omega}_t) := \lambda^{\text{pre}} \cdot \text{Vec}(\mathbf{B}_t \odot \mathbf{P}^{\text{src}}), \tag{15}$$

where $\text{Diag}(\cdot)$ refers to the vectorized diagonal elements of the input matrix while $\text{Vec}(\cdot)$ is an operator to reshape a matrix into a vector. In Eq. (15), $\mathbf{B}_t \in \mathbb{R}^{H \times W}$ is a binary matrix dependent on $t$ and $\mathbf{P}^{\text{src}}$, which is heuristically defined as

$$\mathbf{B}_t[h, w] := \mathbb{I}\left[\mathbf{P}^{\text{src}}[h, w] \geq 1 - \cos\left(\frac{\pi(T - t)}{T\delta}\right)\right], \tag{16}$$

where $\mathbb{I}[\cdot]$ is the indicator function. Note that $\delta$ is a hyperparameter that controls the period of the cosine function and is simply set to $1.5$ in our implementation, which makes the cosine function monotonically decreasing. The binary mask $\mathbf{B}_t$ allows us to ignore the error between the true mean of the posterior $p(\hat{\mathbf{x}}_t^{\text{src}}|\mathbf{x}_t^{\text{tgt}}, \mathbf{y}^{\text{tgt}})$ and its estimate $\mathbf{x}_t^{\text{tgt}}$ in foreground regions. As visualized in Figure 4, $\mathbf{B}_t$ encourages the editable foreground area to be larger as the time step $t$ gets closer to 0.

### 4.5 Update Equation

The new reverse process based on the proposed conditional score in Eq. (10) is given by

$$\mathbf{x}_{t-1}^{\text{tgt}} \approx \frac{\sqrt{\alpha_{t-1}}}{\sqrt{\alpha_t}}\mathbf{x}_t^{\text{tgt}} + (1 - \alpha_t)\gamma_t\left(\nabla_{\mathbf{x}_t^{\text{tgt}}}\log p(\mathbf{x}_t^{\text{tgt}}|\mathbf{y}^{\text{tgt}}) + \nabla_{\mathbf{x}_t^{\text{tgt}}}\log p(\hat{\mathbf{x}}_t^{\text{src}}|\mathbf{x}_t^{\text{tgt}}, \mathbf{y}^{\text{tgt}})\right), \tag{17}$$

where the additional guidance term comes from the replacement of the standard score function in Eq. (5). Finally, the equation for our reverse process is derived by using Eq. (14) and converting to a similar form as Eq. (3) as

$$\mathbf{x}_{t-1}^{\text{tgt}} = \sqrt{\alpha_{t-1}}\hat{f}_\theta(\mathbf{x}_t^{\text{tgt}}, t, \mathbf{y}^{\text{tgt}}) + \sqrt{1 - \alpha_{t-1}}\hat{\epsilon}_\theta(\mathbf{x}_t^{\text{tgt}}, t, \mathbf{y}^{\text{tgt}}) - \gamma_t\boldsymbol{\Omega}_t(\mathbf{x}_t^{\text{tgt}} - \hat{\mathbf{x}}_t^{\text{src}}), \tag{18}$$

where the noise prediction $\epsilon_\theta(\mathbf{x}_t^{\text{tgt}}, t, \mathbf{y}^{\text{tgt}})$ is replaced by the enhanced noise estimation $\hat{\epsilon}_\theta(\mathbf{x}_t^{\text{tgt}}, t, \mathbf{y}^{\text{tgt}})$ with the proposed cross-attention mixup. Therefore, compared to the original score function, the proposed conditional guidance adjusts $\mathbf{x}_t^{\text{tgt}}$ towards $\hat{\mathbf{x}}_t^{\text{src}}$ depending on the elements in the precision matrix $\boldsymbol{\Omega}_t$.

## 5 Experiments

This section compares the proposed method, referred to as CSG, with the state-of-the-art methods such as Prompt-to-Prompt [33] and Pix2Pix-Zero [35], along with the simple variant of DDIM described in Section 3.2 on top of the pretrained Stable Diffusion model [26]. We also present ablation study results to analyze the performance of the proposed components.

### 5.1 Implementation Details

Our method is implemented based on the publicly available official code of Pix2Pix-Zero [35] in PyTorch [44] and tested on a single NVIDIA A100 GPU. For faster generation, we adopt 50 forward steps using Eq. (1) and 50 reverse steps for target image generation. We obtain a source prompt by employing the pretrained vision-language model [45] while a target prompt is made by replacing words in the source prompt with the target-specific ones. For example, in the case of the "dog-to-cat" task, if the source prompt is "a cute little white dog", the target prompt is set to "a cute little white cat". For translating target images, all methods employ the classifier-free guidance [46]. For fair comparisons, we run the official codes of Prompt-to-Prompt [33][1] and Pix2Pix-Zero [35][2], where we utilize the same final latent and target prompt when synthesizing target images.

---

[1]https://github.com/google/prompt-to-prompt
[2]https://github.com/pix2pixzero/pix2pix-zero

Table 1: Quantitative comparisons with existing methods [4, 33, 35] relying on the pretrained Stable Diffusion [26], where real images are sampled from LAION 5B dataset [20]. DDIM denotes the simple inference using Eq. (3). Bold-faced numbers in black and red represent the best and second-best performance in each row.

| Task | DDIM [4] | | | Prompt-to-Prompt [33] | | | Pix2Pix-Zero [35] | | | CSG (Ours) | | |
|---|---|---|---|---|---|---|---|---|---|---|---|---|
| | CS ($\uparrow$) | SD ($\downarrow$) | RD ($\downarrow$) | CS ($\uparrow$) | SD ($\downarrow$) | RD ($\downarrow$) | CS ($\uparrow$) | SD ($\downarrow$) | RD ($\downarrow$) | CS ($\uparrow$) | SD ($\downarrow$) | RD ($\downarrow$) |
| cat $\rightarrow$ dog | 0.2921 | 0.0725 | 0.4325 | 0.2992 | 0.0338 | 0.1756 | **0.3015** | **0.0226** | **0.1589** | **0.3014** | **0.0192** | **0.0217** |
| dog $\rightarrow$ cat | 0.2903 | 0.0748 | 0.4608 | **0.2959** | 0.0295 | **0.2085** | 0.2954 | **0.0220** | 0.3229 | **0.2958** | **0.0150** | **0.0192** |
| wolf $\rightarrow$ lion | 0.2990 | 0.0726 | 0.8856 | 0.2934 | 0.0307 | 0.2569 | **0.3014** | **0.0269** | **0.1827** | **0.2999** | **0.0253** | **0.0778** |
| zebra $\rightarrow$ horse | **0.3006** | 0.0933 | 0.8659 | 0.2939 | 0.0423 | 0.7944 | **0.2944** | **0.0212** | **0.1372** | 0.2918 | **0.0189** | **0.1176** |
| dog $\rightarrow$ dog w/ glasses | 0.3139 | 0.0689 | 0.3541 | **0.3250** | 0.0184 | 0.1340 | **0.3247** | **0.0104** | **0.0921** | 0.3240 | **0.0097** | **0.0139** |

Table 2: Contribution of the conditional score guidance and the cross-attention mixup tested on LAION 5B [20]. 'CSG w/o Mixup' synthesizes target images without the cross-attention mixup.

| Task | DDIM [4] | | | CSG w/o Mixup | | | CSG | | |
|---|---|---|---|---|---|---|---|---|---|
| | CS ($\uparrow$) | SD ($\downarrow$) | RD ($\downarrow$) | CS ($\uparrow$) | SD ($\downarrow$) | RD ($\downarrow$) | CS ($\uparrow$) | SD ($\downarrow$) | RD ($\downarrow$) |
| cat $\rightarrow$ dog | 0.2921 | 0.0725 | 0.4325 | **0.3008** | **0.0199** | **0.0229** | **0.3014** | **0.0192** | **0.0217** |
| dog $\rightarrow$ cat | 0.2903 | 0.0748 | 0.4608 | **0.2959** | **0.0161** | **0.0226** | **0.2958** | **0.0150** | **0.0192** |
| wolf $\rightarrow$ lion | 0.2990 | 0.0726 | 0.8856 | **0.3000** | **0.0273** | **0.0829** | **0.2999** | **0.0253** | **0.0778** |
| zebra $\rightarrow$ horse | **0.3006** | 0.0933 | 0.8659 | 0.2916 | **0.0195** | **0.1231** | **0.2918** | **0.0189** | **0.1176** |
| dog $\rightarrow$ dog w/ glasses | 0.3139 | 0.0689 | 0.3541 | **0.3238** | **0.0103** | **0.0654** | **0.3240** | **0.0097** | **0.0139** |

## 5.2 Evaluation Metrics

To compare the proposed method with previous state-of-the-art techniques, we employ two evaluation metrics widely used in existing studies: CLIP-similarity (CS) [47] and Structure Distance (SD) [48]. CS assesses the alignment between the target image and the target prompt while SD estimates the overall structural disparity between the source and target images.

Furthermore, we introduce a novel metric referred to as Relational Distance (RD), which quantifies how faithfully the relational information between source images is preserved between translated target images. With an ideal image-to-image translation algorithm, the distance between two source images should have a strong correlation with the distance between their corresponding target images. Note that this metric has proven successful in the context of knowledge distillation [49]. To compute the distance between two images, we adopt the perceptual loss [50]. In Section A.1, we provide a detailed description of RD. For quantitative evaluation, we select 250 images with the highest CLIP similarity from the LAION 5B dataset [20].

## 5.3 Quantitative Results

Table 1 presents the quantitative results in comparison with the state-of-the-art methods [33, 35] and the naïve DDIM [4] inference using Eq. (3). As presented in the table, CSG is always the best in terms of SD and RD while it is outperformed by other methods in terms of CS. Although existing approaches have higher values of CS, our algorithm is more effective for preserving the structure. In addition, CSG is more efficient than Pix2Pix-Zero in terms of speed and memory usage since we do not need to perform the backpropagation through the noise prediction network.

## 5.4 Qualitative Results

Figure 5 visualizes images generated by reconstruction, the naïve DDIM method using Eq. (3), Prompt-to-Prompt [33], Pix2Pix-Zero [35], and CSG. As illustrated in the figure, CSG successfully edits the content effectively and preserves the background properly while all other methods often fail to preserve the structural information of the primary objects. In addition to the real examples, we present additional qualitative results using synthesized images by Stable Diffusion in Figure 6, which also demonstrates that CSG achieves outstanding performance.

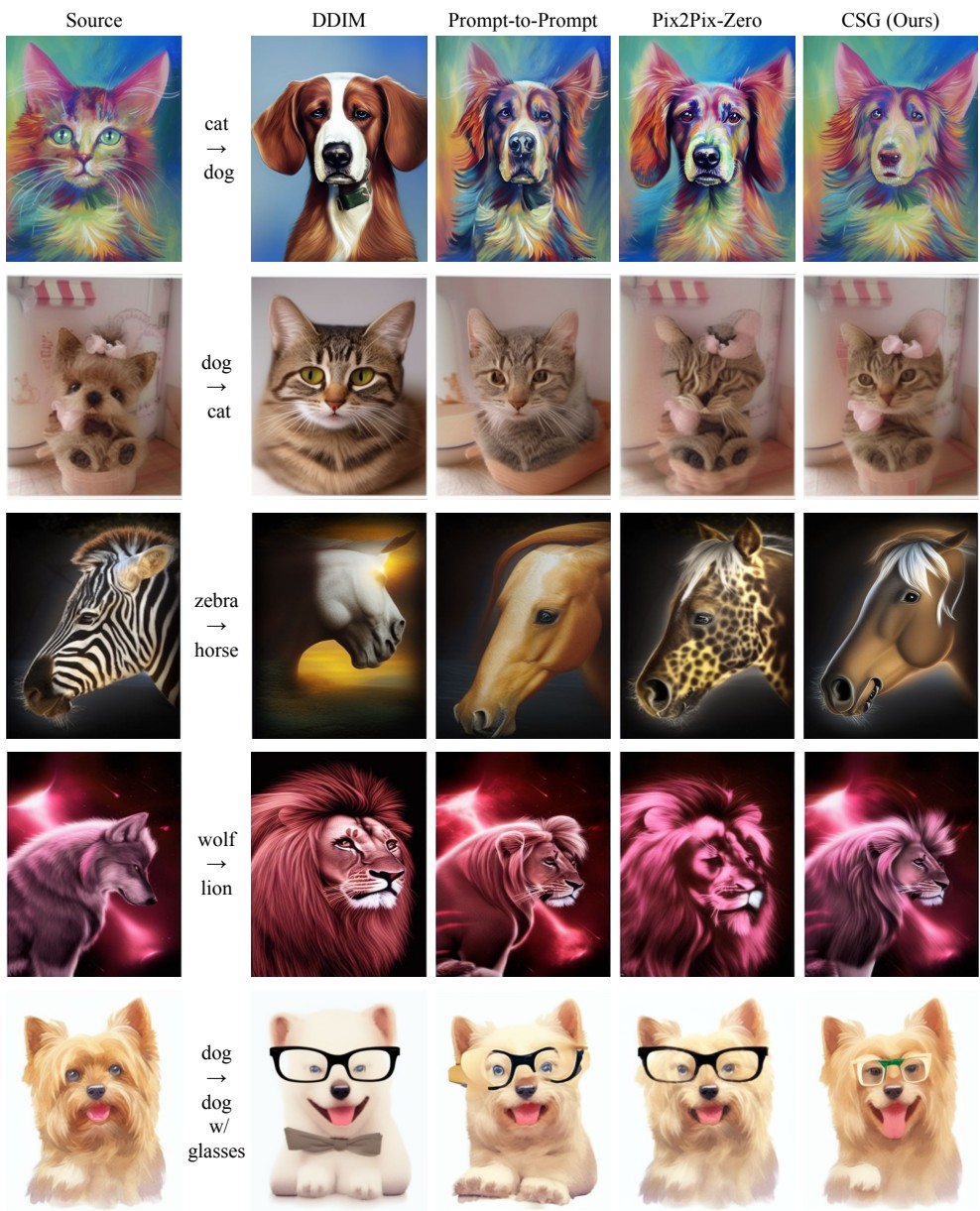

Figure 5: Qualitative comparisons between CSG and other methods tested with the real images sampled from LAION 5B dataset [20]. CSG produces translated images with higher-fidelity.

## 5.5 Ablation Study

Table 2 presents the results from the ablation study that analyzes the effect of the proposed components on various tasks when implemented on top of the Stable Diffusion model. The results imply that the conditional score guidance without the cross-attention mixup in Eq. (13), denoted by 'CSG w/o Mixup', is still helpful, and the cross-attention mixup further enhances text-driven image editing performance when combined with the conditional score guidance.

## 6 Conclusion

We presented a training-free text-driven image translation method using a pretrained text-to-image diffusion model. Different from existing methods relying on a simple score conditioned only on

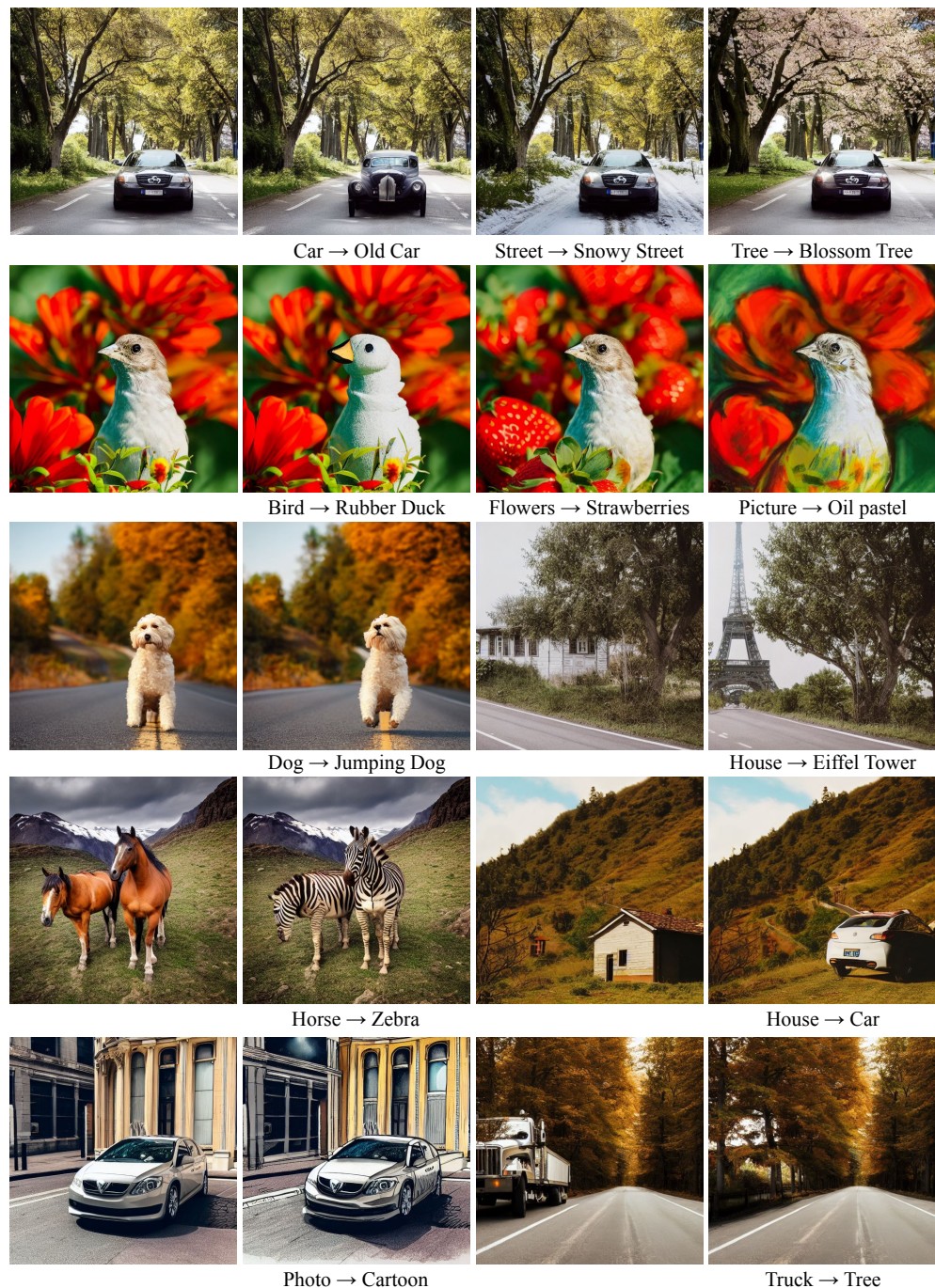

Figure 6: Qualitative results of the proposed method on the synthesized images by the pretrained Stable Diffusion [26].

a target textual input, we formulated a conditional score conditioned on both a source image and a source text in addition to the target prompt. To compute the additional guiding term in our novel conditional score function, we assumed a Gaussian distribution for the posterior distribution, where its mean is simply set to the target latent while the covariance matrix is estimated based on the background mask. For a more accurate estimation of the conditional score function, we incorporated a new cross-attention mixup. Experimental results show that the proposed method achieves outstanding performance in various text-driven image-to-image translation scenarios.

**Acknowledgments** This work was partly supported by LG AI Research. It was also partly supported by the Bio & Medical Technology Development Program of the National Research Foundation (NRF) [No. 2021M3A9E4080782] and Institute of Information & communications Technology Planning & Evaluation (IITP) grant [No. 2022-0-00959, No. 2021-0-01343, No. 2021-0-02068], funded by the Korea government (MSIT).

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

# A    Appendix

In this appendix, we first formulate our introduced metric regarding relational distance (RD). Then, we present additional quantitative results to show the effectiveness of the proposed method by measuring the background difference between source and target images using Learned Perceptual Image Patch Similarity metric [51] referred to as BG-LPIPS. Also, we demonstrate additional qualitative results of CSG compared with the state-of-the-art methods [4, 33, 35]. Finally, we discuss the limitations and potential negative societal impacts.

## A.1    Relational Distance

Relational distance (RD) is introduced to measure how faithfully the relational information between source images is preserved between synthesized target images, which is given by

$$\text{RD} = \min_{\gamma} \frac{1}{n}\|G^{\text{tgt}} - \gamma G^{\text{src}}\|_F^2, \tag{19}$$

where $\|\cdot\|_F$ denotes the Frobenius norm. In the above equation, $G^{\text{tgt}}$ and $G^{\text{src}}$ are $n \times n$ matrices, where the entry $G^{\text{tgt}}[i, j]$ in the $i^{\text{th}}$ row and $j^{\text{th}}$ column of $G^{\text{tgt}}$ denotes the perceptual distance [50] between the $i^{\text{th}}$ and $j^{\text{th}}$ target images while $G^{\text{src}}[i, j]$ represents the perceptual distance between the $i^{\text{th}}$ and $j^{\text{th}}$ source images.

## A.2    Additional Quantitative Results

In addition to Table 1 and 2, which utilize CLIP-similarity [47], structure distance [48], and relational distance for evaluation, we also report BG-LPIPS scores of the proposed method and existing frameworks [4, 33, 35], where BG-LPIPS aims to measure the background difference between source and target images based on the LPIPS metric [51]. As presented in Table 3, CSG always achieves the lowest BG-LPIPS scores, implying that the proposed method more effectively preserves the background region. Moreover, Table 4 demonstrates the superiority of our conditional guidance and cross-attention mixup strategy.

Table 3: Additional quantitative results to compare the proposed method with text-driven image-to-image translation methods [4, 33, 35] using the pretrained Stable Diffusion [26], where real images are sampled from LAION 5B dataset [20]. DDIM denotes the simple inference using Eq. 18. Black and red bold-faced numbers represent the best and second-best performance in each row.

| Task | DDIM [4] | Prompt-to-Prompt [33] | Pix2Pix-Zero [35] | CSG (Ours) |
|------|----------|-----------------------|-------------------|------------|
|      | BG-LPIPS ($\downarrow$) | BG-LPIPS ($\downarrow$) | BG-LPIPS ($\downarrow$) | BG-LPIPS ($\downarrow$) |
| cat $\rightarrow$ dog | 0.3834 | 0.2502 | **0.2111** | **0.1867** |
| dog $\rightarrow$ cat | 0.3602 | 0.2333 | **0.1983** | **0.1645** |
| wolf $\rightarrow$ lion | 0.4042 | 0.2852 | **0.2402** | **0.2384** |
| zebra $\rightarrow$ horse | 0.4127 | 0.3162 | **0.2312** | **0.2303** |

Table 4: Additional ablation study results from LAION 5B dataset [20] to analyze the effectiveness of the conditional score guidance and the cross-attention mixup. 'CSG w/o Mixup' synthesizes target images without using the cross-attention mixup.

| Task | DDIM [4] | CSG w/o Mixup | CSG |
|------|----------|---------------|-----|
|      | BG-LPIPS ($\downarrow$) | BG-LPIPS ($\downarrow$) | BG-LPIPS ($\downarrow$) |
| cat $\rightarrow$ dog | 0.3834 | **0.1896** | **0.1867** |
| dog $\rightarrow$ cat | 0.3602 | **0.1682** | **0.1645** |
| wolf $\rightarrow$ lion | 0.4042 | **0.2510** | **0.2384** |
| zebra $\rightarrow$ horse | 0.4127 | **0.2355** | **0.2303** |

## A.3    Additional Qualitative Results

We provide additional qualitative results using real images sampled from the LAION 5B dataset [20] in Figure 7, 8, 9, 10, and 11, which imply the superiority of CSG compared with the state-of-the-art

methods [4, 33, 35]. Additionally, Figure 12, 13, and 14 demonstrate that the proposed method achieves outstanding performance on synthesized images which are given by the pretrained Stable Diffusion [26]. Furthermore, we visualize additional qualitative results in Figure 15 to compare CSG with Pix2Pix-Zero using the synthesized samples, which illustrates that CSG outperforms Pix2Pix-Zero. Unlike CSG, Pix2Pix-Zero struggles with dissimilar object-centric tasks such as house-to-Eiffel tower as presented in the figure since Pix2Pix-Zero enforces shape matching through cross-attention layers.

## A.4  Limitations and Potential Negative Societal Impacts

Our method may fail to edit images with complex prompts due to the incompetence of pretrained text-to-image diffusion models. Similar to other text-driven image-to-image translation methods, our approach also encounters a limitation that it cannot be applied to complex tasks, such as enlarging parts of an object or moving the object, where it would be an interesting future work to tackle these challenges. Regarding potential negative social impacts, our method can generate harmful or misleading contents due to the pretrained model.

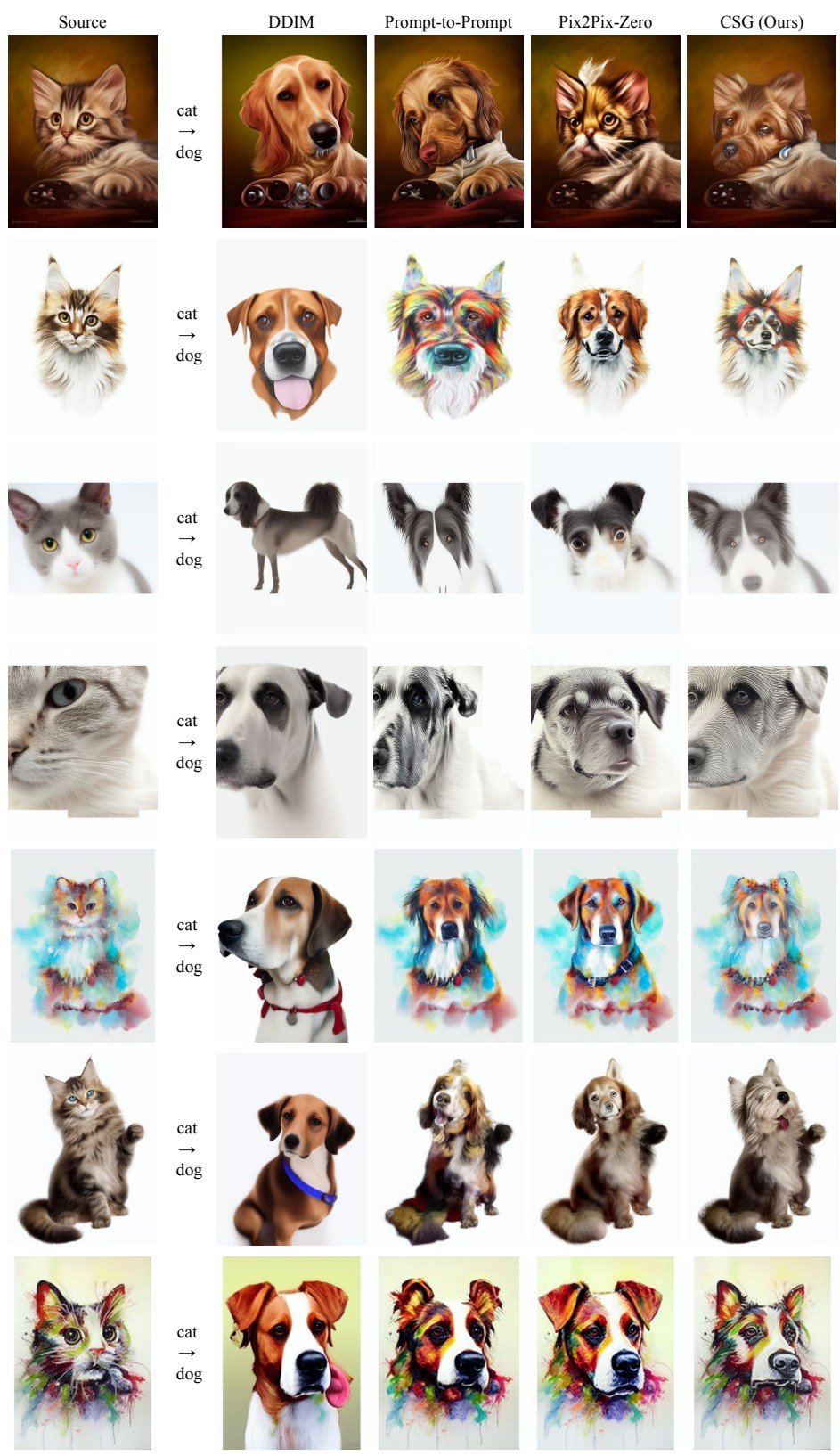

Figure 7: Additional qualitative comparisons between CSG and other state-of-the-art methods tested with the real images sampled from LAION 5B dataset [20] on the cat-to-dog task.

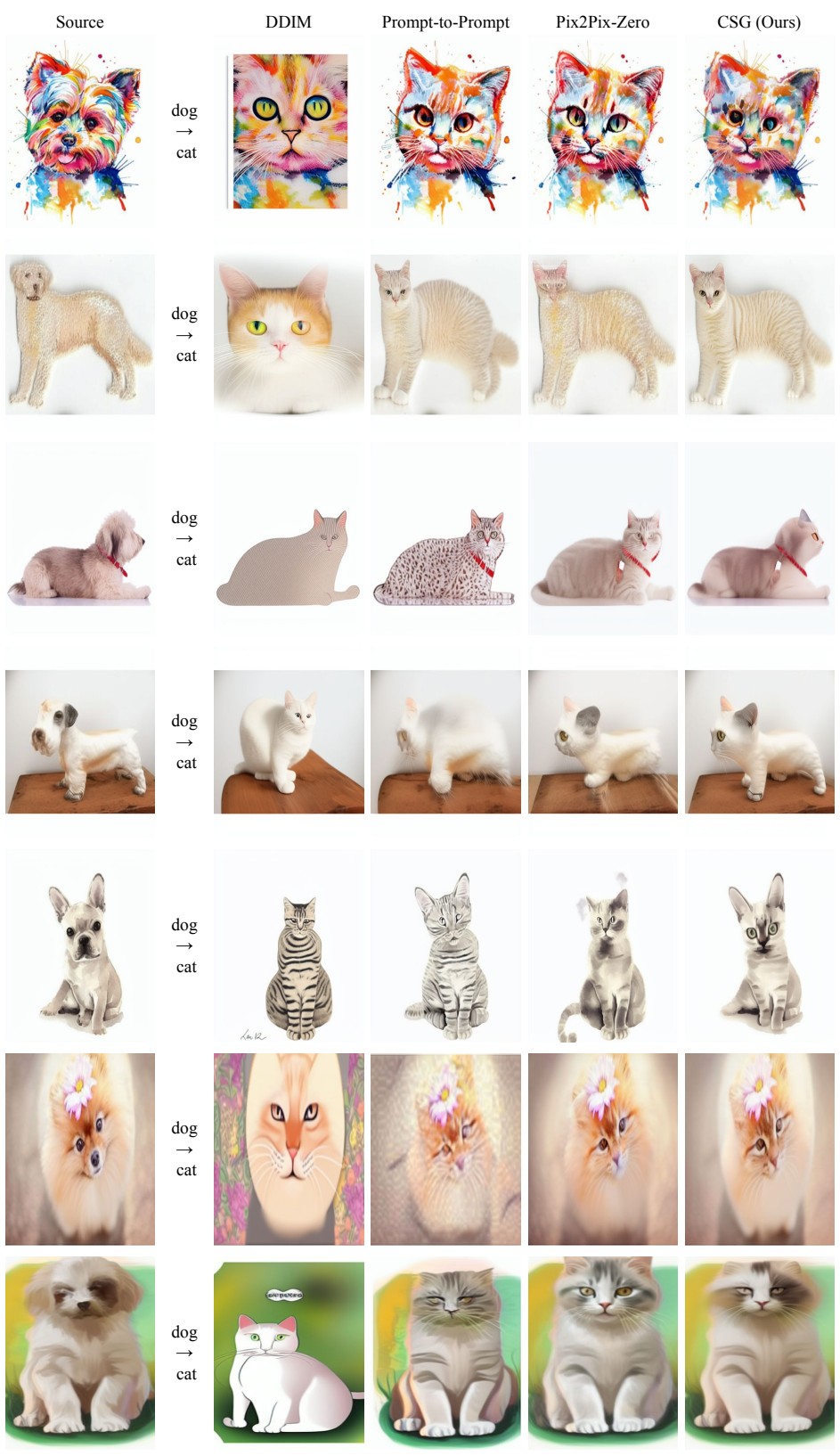

Figure 8: Additional qualitative comparisons between CSG and other state-of-the-art methods tested with the real images sampled from LAION 5B dataset [20] on the dog-to-cat task.

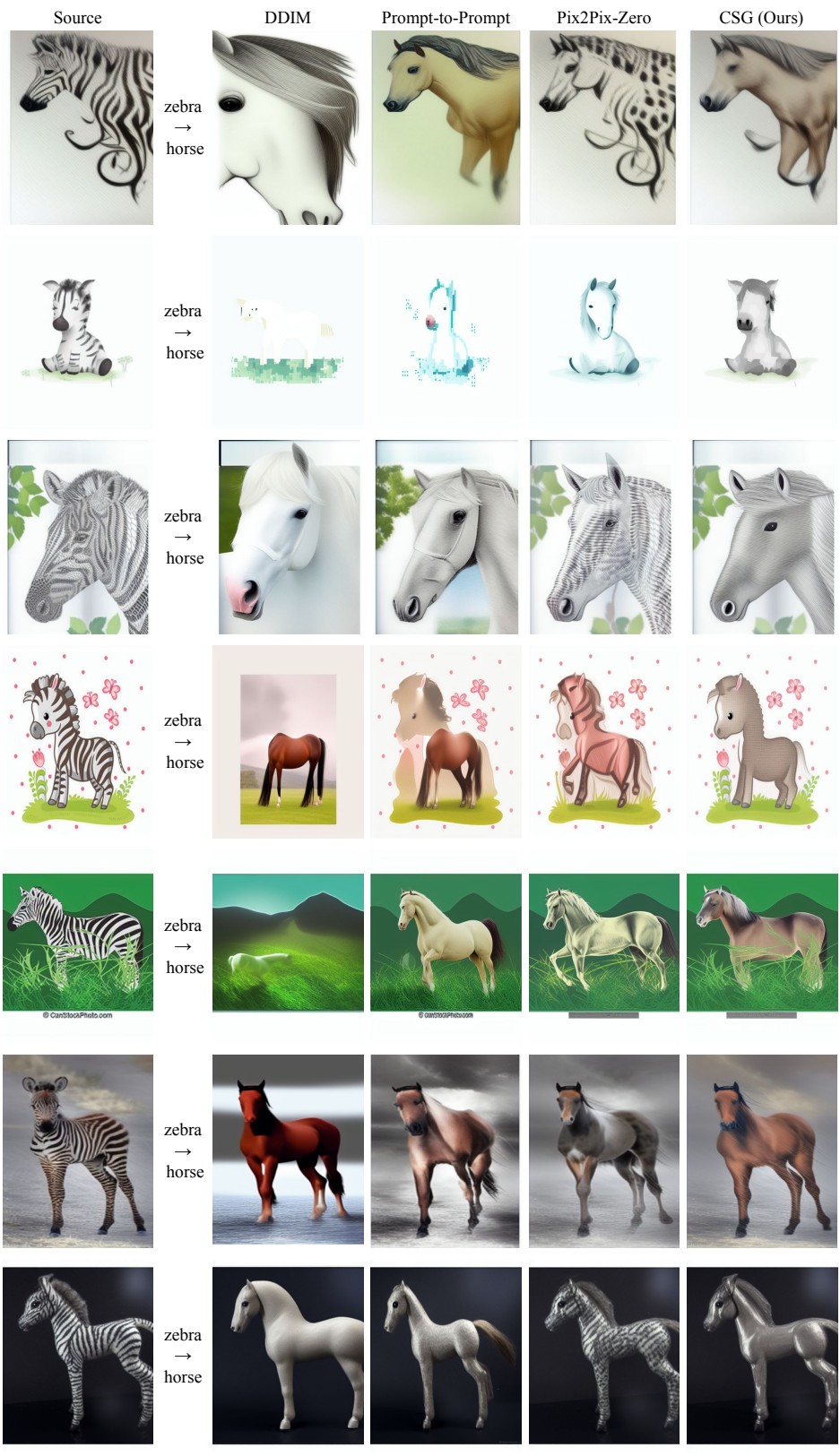

Figure 9: Additional qualitative comparisons between CSG and other state-of-the-art methods tested with the real images sampled from LAION 5B dataset [20] on the zebra-to-horse task.

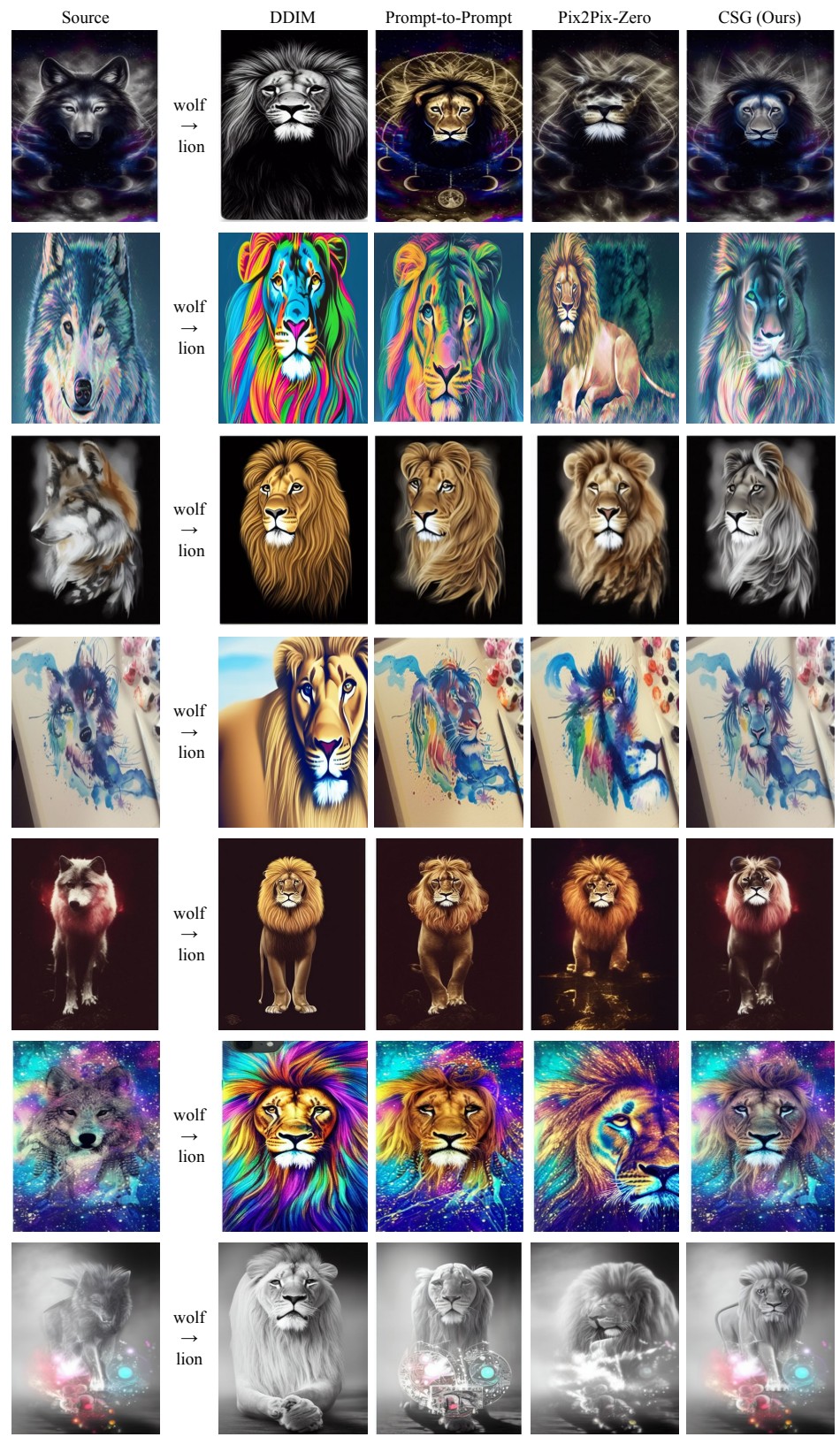

Figure 10: Additional qualitative comparisons between CSG and other state-of-the-art methods tested with the real images sampled from LAION 5B dataset [20] on the wolf-to-lion task.

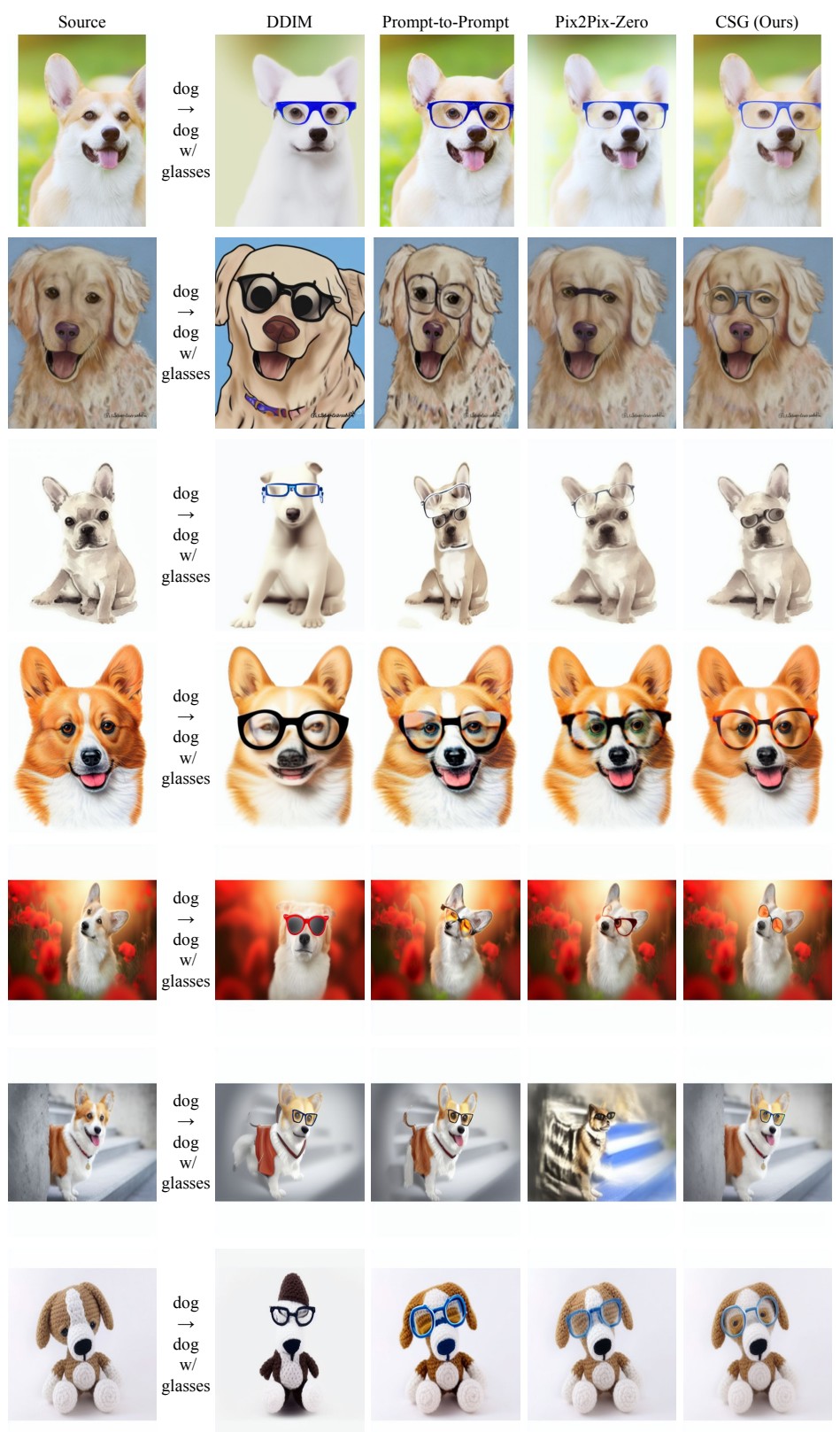

Figure 11: Additional qualitative comparisons between CSG and other state-of-the-art methods tested with the real images sampled from LAION 5B dataset [20] on the dog-to-dog with glasses task.

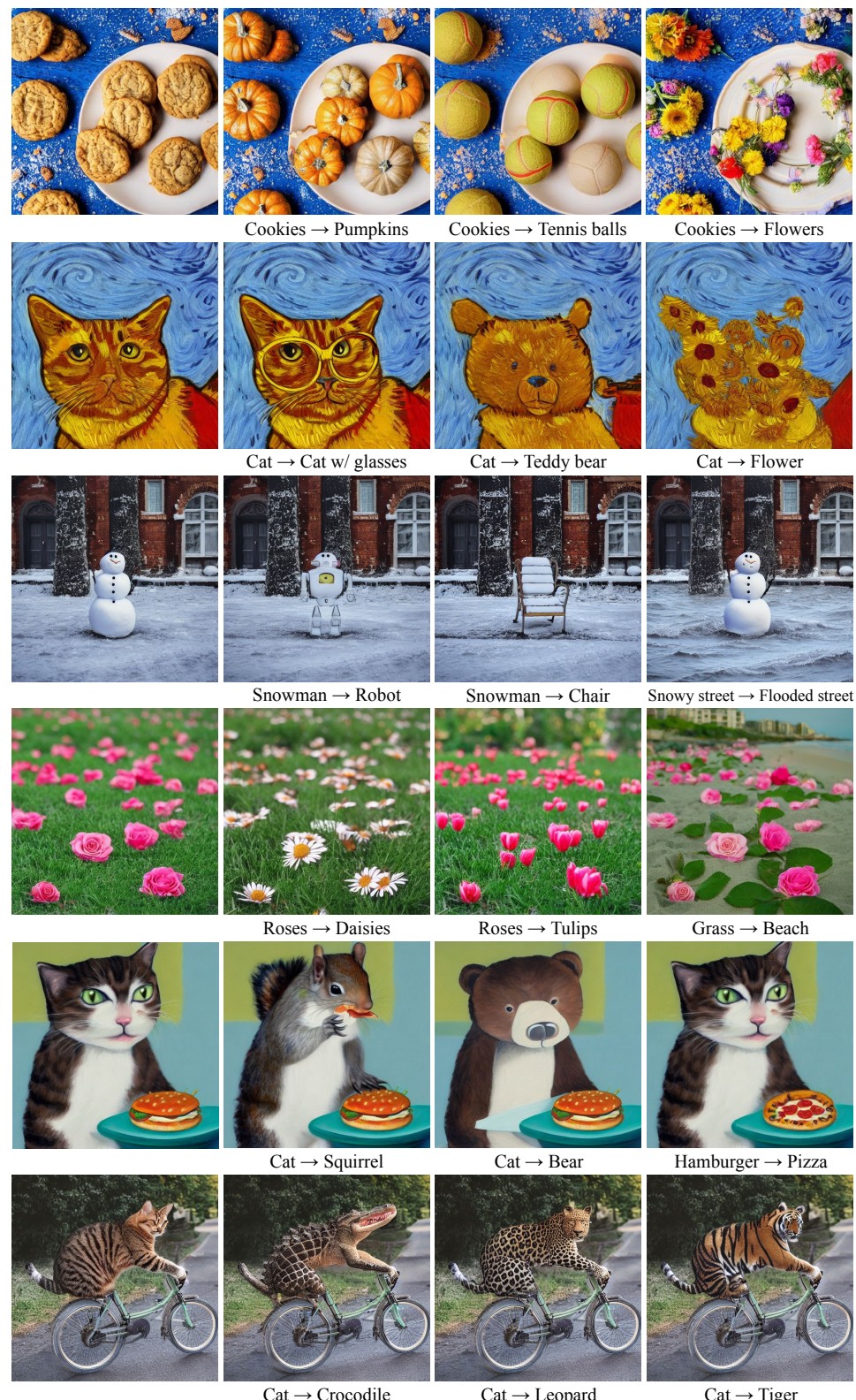

Figure 12: Additional qualitative results of the proposed method on the synthesized images by the pretrained Stable Diffusion [26].

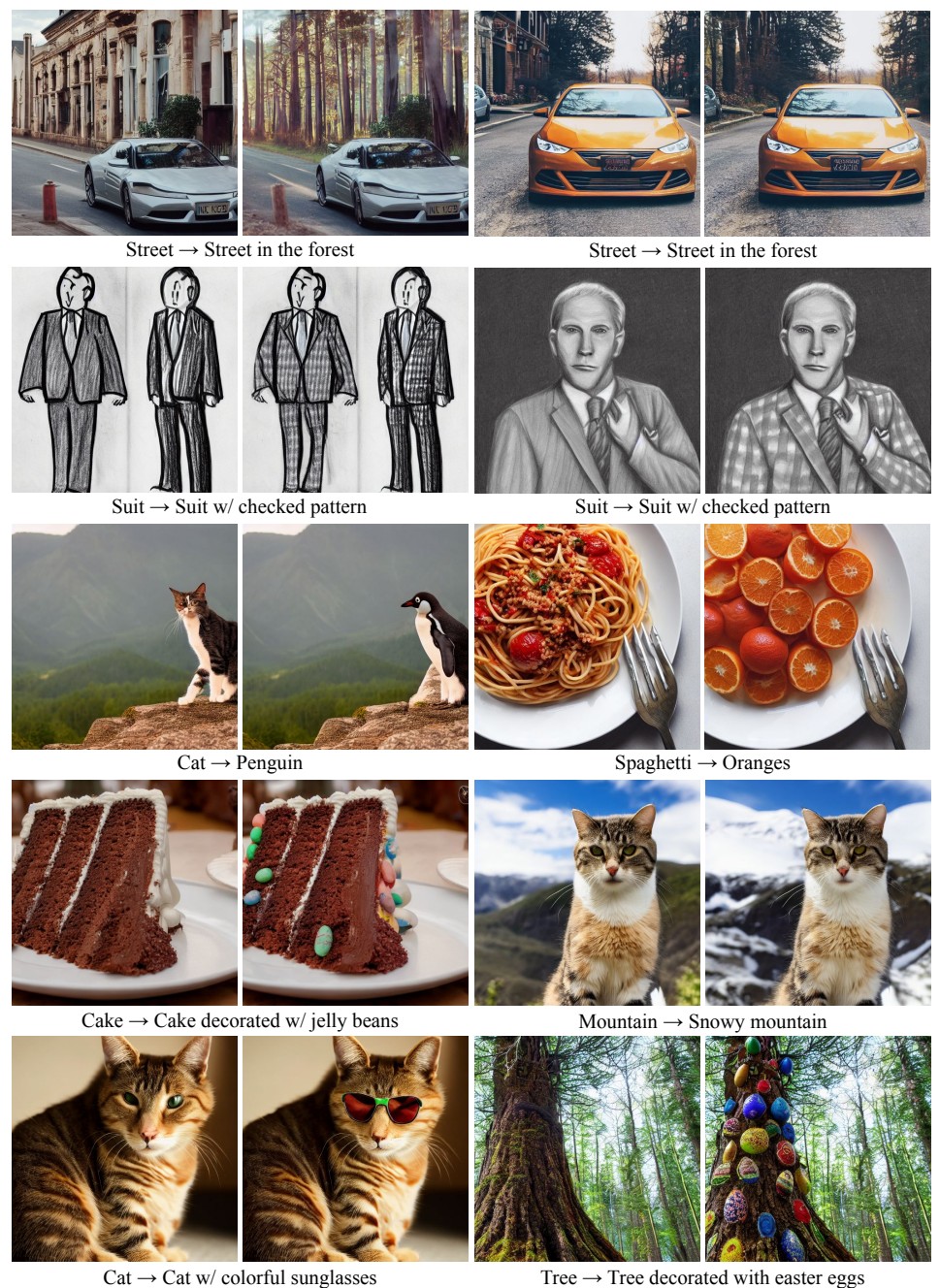

Figure 13: Additional qualitative results of the proposed method on the synthesized images by the pretrained Stable Diffusion [26].

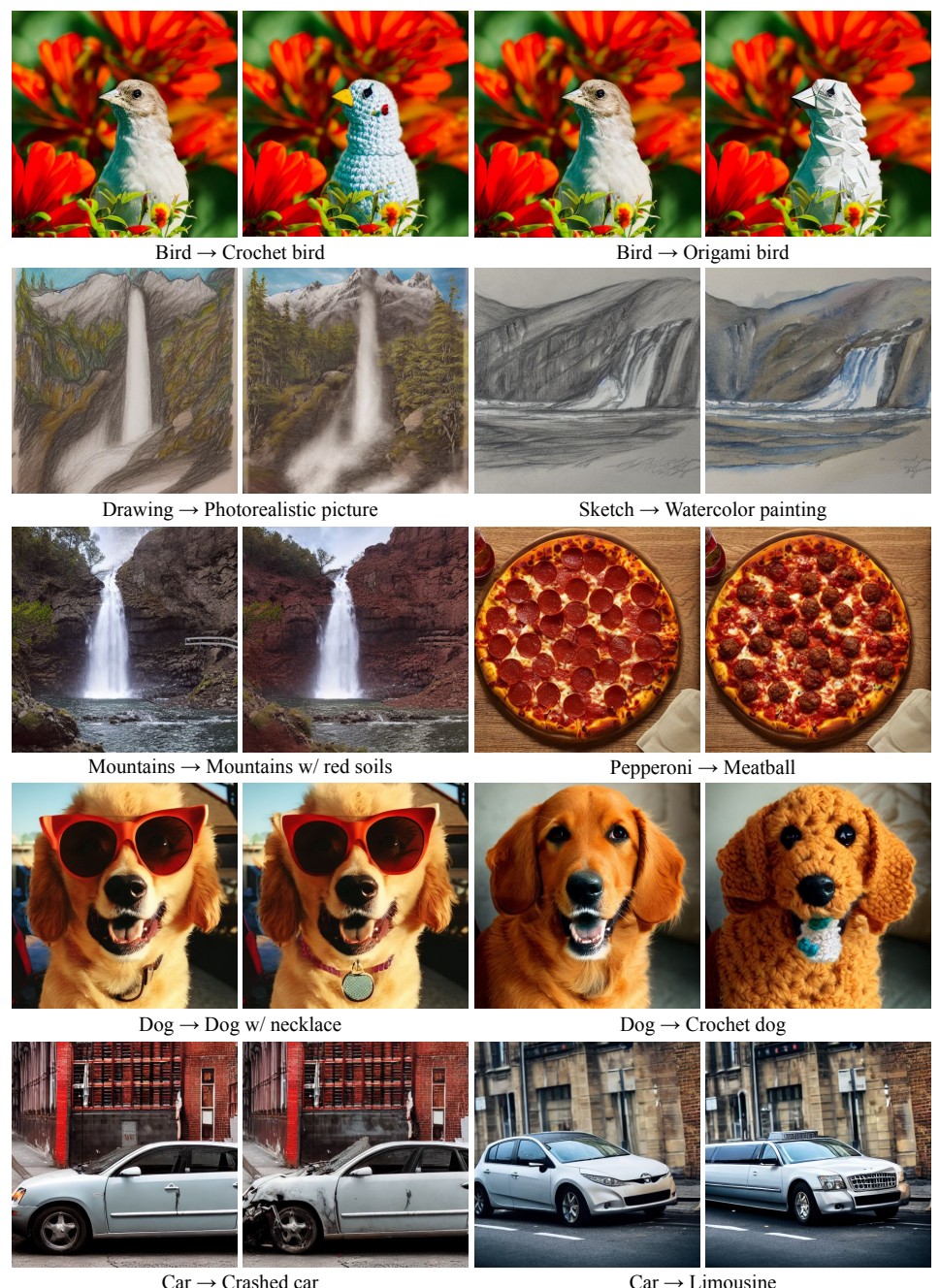

Bird → Crochet bird      Bird → Origami bird

Drawing → Photorealistic picture      Sketch → Watercolor painting

Mountains → Mountains w/ red soils      Pepperoni → Meatball

Dog → Dog w/ necklace      Dog → Crochet dog

Car → Crashed car      Car → Limousine

Figure 14: Additional qualitative results of the proposed method on the synthesized images by the pretrained Stable Diffusion [26].

Source       Pix2Pix-Zero       CSG

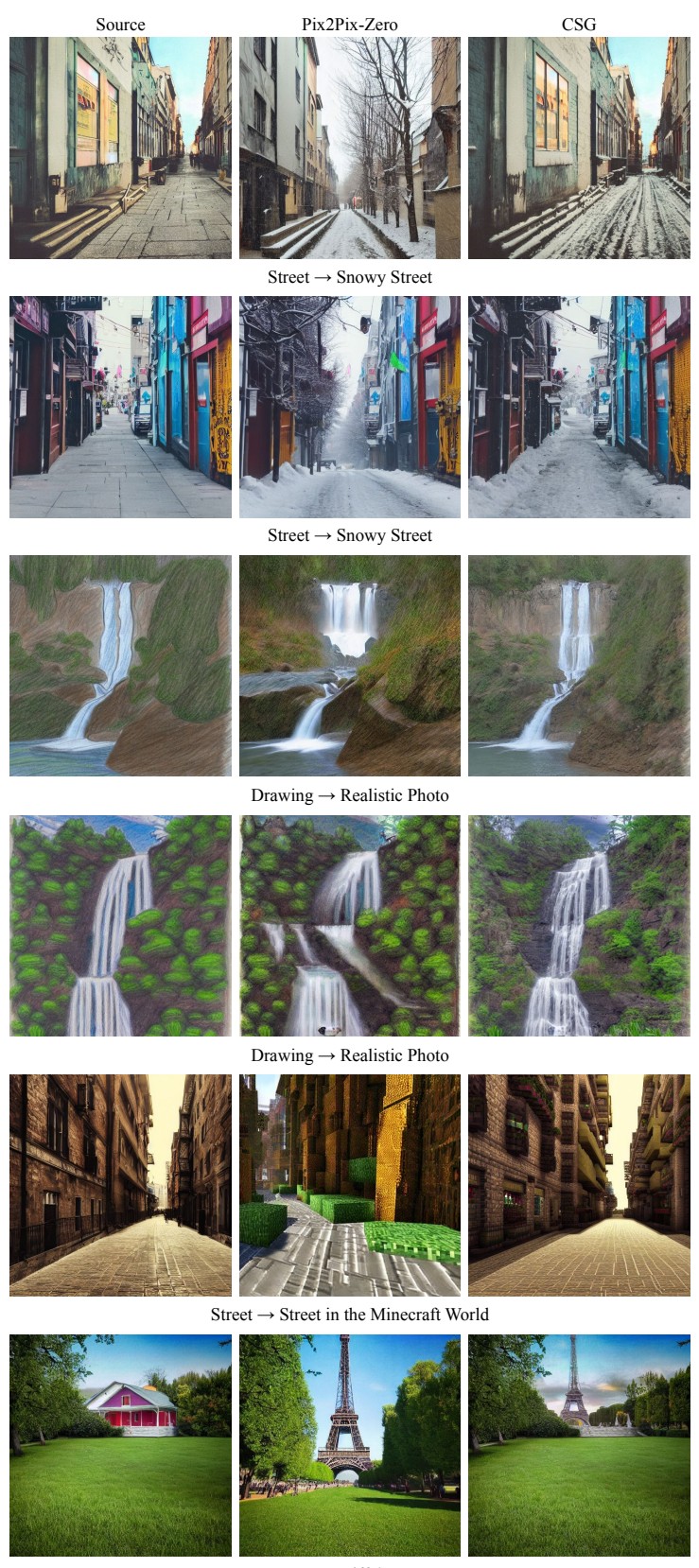

Street → Snowy Street

Street → Snowy Street

Drawing → Realistic Photo

Drawing → Realistic Photo

Street → Street in the Minecraft World

House → Eiffel Tower

Figure 15: Qualitative comparisons between CSG and Pix2Pix-Zero on the synthesized images by the pretrained Stable Diffusion [26].

