# OpenReview forum: "Conditional Score Guidance for Text-Driven Image-to-Image Translation"
_NeurIPS.cc/2023/Conference — NeurIPS 2023 poster_

### Official Review · Reviewer_rP2j · 2023-06-30

**Soundness:** 3 good
**Presentation:** 3 good
**Contribution:** 3 good
**Rating:** 6
**Confidence:** 3

**Summary:**

This paper introduces a novel approach for text-driven image-to-image translation tasks. The main contribution of this work is the development of a conditional score function that takes into account both the source image and text, in addition to the standard condition with the target text. The new score function consists of two terms: the standard score function conditioned on the target prompt, and the guiding score function, which models the posterior of the source latent given the target latent and target prompt. The paper also proposes an effective mixup strategy in cross-attention layers of the text-to-image diffusion model to facilitate image-to-image translation.  Experimental results that demonstrate the outstanding performance of the proposed method on various tasks. The paper also introduces an intuitive performance evaluation metric that measures the fidelity of pairwise relations between images before and after translation.


**Strengths:**

1: Detailed theoretical explanation for the algorithm used and the overall setup, providing a strong foundation for the study.

2: The paper offers a novel mixup method that enhances the conditional score guidance, and the mixup method effectively combines two outputs of cross-attention layers.

3: Comprehensive experimentation and qualitative/quantitative measures back up the claims. Comparison with state-of-the-art methods and showcased superiority in most cases.

4: The paper provides a novel metric (Relational Distance) for evaluating the methodology. The new metric quantifies how faithfully the relational information between source images is preserved between translated target image, which is a good contribution.

**Weaknesses:**

1: Lack of clarity on the practicality of the proposed method. It would be very valuable to discuss the computational cost (time, memory) and efficiency of the proposed method compared to other techniques.

2: The paper only focuses on one pre-trained model (Stable Diffusion) for the experiments. More experiments regarding other pre-trained methods should be shown to show the proposed method generalize well.

3:  Insufficient coverage of the limitations of the method. There is no extensive discussion on the limitations of their method and what scenarios it won't work well in

**Questions:**

The method proposed appears to be heavily reliant on the quality of the source image. Could there be issues if the source image isn't high quality, for instance, leading to degrading the results?

Please see the weaknesses for more. I will change my rating based on the rebuttal and other reviewers' comments.


**Limitations:**

Limitations are not discussed. One potential limitation is listed in the questions.

---

> ### Author Rebuttal · Authors · 2023-08-10
>
> We truly thank you for your constructive and positive comments and below are our responses to the main questions.
>
> Q1. Computational cost in terms of time and memory
>
> In order to observe the realistic speed of each algorithm, we measure the wall clock time using a NVIDIA A100 GPU with a single image while checking the memory consumption.
> Although the naive DDIM translation algorithm retains the fastest inference time as presented in Table A8, it achieves poor generation results as mentioned in the main paper and supplementary material.
> For the theoretical inference comparison, note that CSG approximately requires an extra 0.5x inference cost of DDIM since CSG needs an extra computation of reversing the latent using the source prompt embedding different from DDIM.
> However, in case of CSG and Pix2Pix-Zero, there is a disparity between theoretical and practical inference costs since the communication cost for copying cross-attention maps from the GPU memory to the CPU memory is not negligible.
> In CSG, in order to save the GPU memory, our algorithm applies the resizing operation in CPU of the cross-attention maps for computing the smooth content mask, which further makes the inference speed become slower.
> Therefore, the practical speed can be reduced if we have enough GPU memory.
>
>
>
>
>
> Table A8: Computational cost of the proposed method compared to DDIM and Pix2Pix-Zero.
>
> | | DDIM  | Pix2Pix-Zero  | CSG w/o mixup | CSG |
> |:----------------:|:---------------:|:---------------:|:---------------:|:---------------:|
> |    time/image (s)  | 5.129  | 28.647 | 19.791 | 25.736 |
> |    GPU Memory (GB)   | 6.840  | 11.546 | 10.030 | 10.040 |
>
>
>
> Q2. Other pre-trained text-to-image diffusion models
>
> Our framework generalizes well to the diffusion models. To show that the proposed method generalizes well, we visualize generated images given by CSG using LDM in Figure G of the rebuttal document (Rebuttal_CSG.pdf).
> The figure demonstrates that our conditional guidance method can generalize well.
> Note that LDM is similar to Stable Diffusion, however the training data of LDM and its resolution are different from that of Stable Diffusion.
> In case of other pretrained text-to-image diffusion models such as DALLE-2 [E1] and Imagen [E2], they are not publicly available, so we can not test CSG using the pre-trained text-to-image diffusion models.
>
>
>
> Q3. Limitations
>
> Our method can fail to edit images with complex prompts due to the incompetence of pre-trained text-to-image diffusion models. As other text-driven image-to-image translation methods, the proposed method also has another limitation that it can not be applied to complex tasks such as enlarging the object or moving the object, where it would be an interesting work to solve the difficult tasks. For the potential negative social impact, our method can generate harmful or misleading contents due to the pre-trained model. We will add the limitations and negative social impacts in the final version.
>
>
>
> Q4. Translation results of degraded source images
>
> We visualize qualitative results using degraded source images in terms of bounding box removal in the object, lighting changes, and noise addition in Figure F in the rebuttal document (Rebuttal_CSG.pdf).
> Although the quality depends on the quality of the source image as presented in the figure, we can enhance the visual quality.
> For example, in case of the removal in the object, we can address the problem by applying existing image inpainting techniques to the source images and then translate the modified source images using CSG.
> We appreciate a good suggestion, and we will discuss the limitation in the final version.
>
>
> Reference
>
> [E1] A. Ramesh et al., Hierarchical Text-Conditional Image Generation with CLIP Latents
> , arXiv 2022.
>
>
> [E2] C. Saharia et al., Photorealistic Text-to-Image Diffusion Models with Deep Language Understanding, NeurIPS 2022.

---

> > ### Comment · Reviewer_rP2j · 2023-08-11
> > **Thanks for the rebuttal**
> >
> > Thank you for crafting the rebuttal!
> >
> > W1: The efficiency of the proposed method is commendable. It's encouraging to see.
> >
> > W2: It would be intriguing to explore the applicability of the proposed method to diffusion models without relying on an auto-encoder for latent generation. Specifically, investigating denoising within the pixel-space. I recognize that this lies beyond the scope of the current approach, so it is entirely fine.
> >
> > W3: The discussion regarding limitations is well-articulated.
> >
> > W4: Your inclusion of visualizations is greatly appreciated. A valuable addition indeed.
> >
> > Overall, I find that my concerns have been thoughtfully addressed. I am inclined to maintain my rating as a weak accept.

---

> > > ### Author Response · Authors · 2023-08-11
> > > **Thanks for the comment**
> > >
> > > We greatly appreciate your positive comments about the visualization about degraded images and the efficiency of the proposed method. In case of the pre-trained text-to-image diffusion models modeling the pixel space, we hope that such models are publicly available so that we can test CSG using the diffusion models. We believe that our method can be incorporated into such models, and it would be intriguing to work towards this direction.
> > >
> > > Overall, we will add limitations and negative social impacts, and revise the main paper to reflect your comments. Once again, we sincerely thank you for your time and efforts to review our paper.
> > >
> > > Best wishes,
> > >
> > > Authors

---

### Official Review · Reviewer_ZGMV · 2023-07-06

**Soundness:** 3 good
**Presentation:** 3 good
**Contribution:** 2 fair
**Rating:** 5
**Confidence:** 3

**Summary:**

This paper propose Conditional Score Guidance (CSG), where the goal is text-driven image-to-image translation by preserving the original context of a source image. They propose two novel components to achieve this: first, conditional score guidance that computes the score based on the combination of text-conditional score and posterior given from the source latent given the target latent. Here, the posterior is estimated by Gaussian distribution modeling. Second, the propose cross-attention mixup, which enhance the quality of image translation. The proposed method demonstrates high-quality compared to various baselines.

**Strengths:**

This paper propose Conditional Score Guidance (CSG), where the goal is text-driven image-to-image translation by preserving the original context of a source image. They propose two novel components to achieve this: first, conditional score guidance that computes the score based on the combination of text-conditional score and posterior given from the source latent given the target latent. Here, the posterior is estimated by Gaussian distribution modeling. Second, the propose cross-attention mixup, which enhance the quality of image translation. The proposed method demonstrates high-quality compared to various baselines.

**Weaknesses:**

Incremental Novelty: Although the paper proposes conditional scores that replace the original scores of a pre-trained text-to-image diffusion model for image-to-image translation, the core methodology appears to be significantly built upon pix2pix-zero. To elevate the novelty, it's recommended that the authors shed more light on how their method extends beyond pix2pix-zero's capabilities. This can be achieved by emphasizing the unique aspects of the approach and providing a clearer delineation of the differences from pix2pix-zero. In addition, regarding performance, the qualitative results provided in the Appendix do not consistently demonstrate superior performance compared to pix2pix-zero. Thus, to substantiate the claimed superiority of the proposed method, it would be beneficial to include additional examples, including those not addressed by pix2pix-zero, to showcase the broader effectiveness of the paper.

Limited Editing Capabilities: The proposed method seems to primarily focus on object-centric editing. The paper could benefit from demonstrating its capabilities for more precise or finer edits to articulate the efficacy of the proposed components, particularly their role in selectively editing the region of interest while retaining irrelevant parts of the image.

Lack of Comparison: The paper should discuss with concurrent work, Delta Denoising Score [1] which was introduced as same purpose as CSG to edit images with minimal modifications. Furthermore, a performance and methodological comparison with Plug-and-Play [2] is highly recommended to provide a more comprehensive understanding of image-to-image translation.
Missing parts: The authors have not provided any discussion on the limitations and potential negative societal impacts of their proposed model, a crucial aspect that is currently missing.

[1] Hertz et al., Delta Denoising Score

[2] Tumanyan et al., Plug-and-Play Diffusion Features for Text-Driven Image-to-Image Translation

**Questions:**

See the weakness part.

**Limitations:**

Since the paper considers image-editing, the author should demonstrate the case when the proposed method fails, e.g., due to the incompetence of pretrained text-to-image diffusion models, the complex prompt might fail, or some attributes that are hard to edit. Also, some ethical warning must be included in the paper to demonstrate the possible misusage of image editing methods.

---

> ### Author Rebuttal · Authors · 2023-08-10
>
> We truly thank you for your constructive and positive comments and below are our responses to the main questions.
>
> Q1. Incremental novelty
>
> Our algorithm is somewhat related to Pix2Pix-Zero in the sense that CSG and Pix2Pix-Zero employ the cross-attention layers in the noise prediction network for text-driven image-to-image translation tasks. However, the two methods are completely different, where CSG uses mixup strategy in cross-attention layers while Pix2Pix-Zero takes a gradient step to reduce the distance between the cross-attention maps given by the reverse process of $\mathrm{x}^{\text{src}}_t$ and $\mathrm{x}^{\text{tgt}}_t$, which requires an additional backpropagation step and leads to slower inference. Moreover, as mentioned by Reviewer fXLA, Reviewer nbbt, and Reviewer rP2j, we propose a principled technique for text-driven image translation based on the conditional score function with reasonable motivations. Such ideas have not been addressed before and are sound both theoretically and empirically, so we believe that our paper has sufficient novelty and contribution.
>
> Q2. Additional comparison with Pix2Pix-Zero
>
> Different from CSG, it is hard to apply Pix2Pix-Zero for dissimilar object-centric tasks such as hose-to-eiffel tower as presented in the right examples of Figure B in the rebuttal document (Rebuttal_CSG.pdf) since Pix2Pix-Zero enforces the shape matching through the cross-attention layers. Also, Figure A-1, A-2, and B visualize additional qualitative examples using CSG and Pix2Pix-Zero, which demonstrate that CSG outperforms Pix2Pix-Zero.
>
> Q3. Limited editing capabilities
>
> We tried to follow the experiment protocol of prior works [Prompt-to-Prompt, Pix2Pix-Zero] by evaluating CSG on object-centric editing tasks. In addition to the object-centric editing tasks, our method can be extended to global editing tasks. To validate the property, we test CSG and Pix2Pix-Zero using global editing tasks, street-to-snowy street and drawing-to-realistic photo tasks. As presented in Table A6, CSG outperforms Pix2Pix-Zero in terms of SD, LPIPS, and RD even with faster translation although Pix2Pix-Zero achieves slightly higher values of CS. In addition to the quantitative results, Figure A-1 and A-2 in the rebuttal document demonstrate that the proposed method achieves better performance on the tasks. For the global editing tasks, note that we replace BG-LPIPS with LPIPS, where LPIPS measures the perceptual similarity using the entire source and target image, which is more suitable for the global editing tasks. We appreciate you for a good suggestion, and we will add the results in the final version.
>
> Table A6: Quantitative results to compare with Pix2Pix-Zero using the pre-trained Stable Diffusion and its synthetic images for Street → Snowy Street and Drawing → Realistic photo tasks. The black bold-faced number represents the best performance for each task in each metric.
>
> | Street → Snowy Street  | CS (↑)  | SD (↓) | LPIPS (↓)  | RD (↓) |
> |:----------------:|:---------------:|:---------------:|:---------------:|:---------------:|
> |    Pix2Pix-Zero  |  **0.3215**  | 0.0186 | 0.2345 | 0.1436 |
> |   CSG    |    0.3125   | **0.0166** | **0.2077** | **0.1340** |
> | **Drawing → Realistic photo** | **CS (↑)**  | **SD (↓)** | **LPIPS (↓)**  | **RD (↓)** |
> |    Pix2Pix-Zero  |  **0.2997**  | 0.0414 | 0.2783 | 0.3933 |
> |   CSG    |    0.2966   | **0.0263** | **0.0722** | **0.2190** |
>
> Q4. Comparison with Plug-and-Play and Delta Denoising Score (DDS)
>
> We present the results of Plug-and-Play in Table A7, where the results of CSG are presented in Table 1 of the main paper. Table 1 and Table A7  imply that the proposed method outperforms Plug-and-Play in most cases. For the dog-to-dog with glasses task, we do not report the BG-LPIPS score, since the background region can be easily preserved for the task. Moreover, we present the qualitative results given by Plug-and-Play in Figure D in the rebuttal document, which contains compatible results with Figure 4 of the main paper. The two figures imply that CSG archives much better qualitative results. Furthermore, considering the qualitative results in Figure C and D, the proposed method even without using the cross-attention mixup outperforms Plug-and-Play. In case of DDS, it is very difficult to perform the suggested experiment during the rebuttal period since the source code of DDS is not publicly available. Note that DDS is still in arXiv only and released only one month before the deadline. We thank you for a comment and we will add the results in the final version.
>
> Table A7: Quantitative results of Plug-and-Play using the pre-trained Stable Diffusion and real images sampled from the LAION 5B for various tasks. The black bold-faced number represents better performance compared with the results of CSG presented in Table 1.
>
> | | CS (↑)  | SD (↓) | BG-LPIPS (↓)  | RD (↓) |
> |:----------------:|:---------------:|:---------------:|:---------------:|:---------------:|
> |    Cat → Dog   | 0.2787  | **0.0107** | **0.1839** | 0.1197 |
> | Dog → Cat | 0.2734  | **0.0102** | 0.2081 | 0.0995 |
> | Wolf → Lion | 0.2776 | 0.0260 | 0.2647 | 0.1410 |
> | Zebra → Horse | 0.2823  | 0.0314 | 0.3223 | 0.5739 |
> | Dog → Dog w/ glasses | 0.2887  | **0.0067** | - | 0.0923 |
>
> Q5. Limitations and potential negative societal impacts
>
> As you mentioned, our method can fail to edit images with complex prompts due to the incompetence of pre-trained text-to-image diffusion models. As other text-driven image-to-image translation methods, the proposed method also has another limitation that it can not be applied to complex tasks such as enlarging the object or moving the object, where it would be an interesting work to solve the difficult tasks. For the potential negative social impact, our method can generate harmful or misleading contents due to the pre-trained model. We will add the limitations and negative social impacts in the final version.

---

> > ### Author Response · Authors · 2023-08-14
> > **After Rebuttal**
> >
> > Dear Reviewer ZGMV,
> >
> > Because the end of discussion period is approaching, we kindly ask you whether our response is helpful to clarify you or not. Also, if you have any questions or additional comments, please do not hesitate to contact us. We thank you for your time and efforts to review our paper.
> >
> > Best wishes,
> >
> > Authors

---

### Official Review · Reviewer_nbbt · 2023-07-08

**Soundness:** 3 good
**Presentation:** 3 good
**Contribution:** 3 good
**Rating:** 5
**Confidence:** 4

**Summary:**

The authors are proposing two sampling techniques in Diffusion Models; Cross Attention Mixup and conditional score guidance.Experiments show that the proposed methods show decent performance compared to baselines.

**Strengths:**

- Qualitative results are interesting
- Reasonable motivations and methods

**Weaknesses:**

Weaknesses are written in the Question part below.

**Questions:**

* (Important) How can we get Eq. 8  from Eq. 7? I understand the second term ($p(x_t^{src}|…)$) in integral ends up being one. My question here is that how can the integral on the first term ($p(x_t^{tgt}|\hat{x}_t^{src}…)$) be removed (or ignored) to get Eq. 8? The simplest way would be to think of it as a constant w.r.t. $x_t^{src}$. However, it is not the case since the conditional (the first term in Eq. 7) would be equal to $p(x_t^{tgt},x_t^{src}|y^{tgt})p(x_t^{src}|y^{tgt})$ by Bayes rule, and it is not a constant w.r.t. $x_t^{src}$.
* (Line 35-36) A bridging paragraph is needed.
* (typo, Algorithm 1) a target prompt embedding.
* It is not intuitive how the covariance $\Omega^{-1}$ is computed, and what role it plays. Can it be visualized? e.g., use only the posterior guidance when do sampling and directly visualize Eq. 14.
* It seems like $\Omega \in \mathbb{R}^{H \times W}$ while diagonal covariance matrix needs to be in $\mathbb{R}^{C \times H \times W}$ (considering the diagonal term only)
* What about sampling speed w/wo the additional guidance term?
* (Important)
    * CSG (wo mixup) evaluation:
        * Classifier free guidance (best weight needs to be searched, e.g., 3-7) v.s. CSG (wo mixup)
    * Ablation:
        * DDIM + mixup v.s. DDIM => to show the effect of mixup


**Limitations:**

- Additional experiments are needed to validate each of proposed method separately.
- Additional comparison between the proposed guidance method (without mixup) and exiting guidance method is needed.
- Visualization of the covariance of the posterior would be helpful to understand the proposed method.
- Additional description on how to get Eq. 8 from Eq. 7.
- It is hard to see the evaluation metric as a novelty since it is not menitoned/analyzed specifically.

Although there are some limitations, I will increase my rating if my questions can be reasonably answered.

---

> ### Author Rebuttal · Authors · 2023-08-10
>
> We truly thank you for your constructive and positive comments and below are our responses to the main questions.
>
> Q1. Mathematical expression of Eq. 8 from Eq. 7
>
> As described in line 164 of the main paper, we get Eq. (8) from Eq. (7) by drawing a sample $\hat{\mathrm{x}}^{\text{src}}_t$ from $p(\mathrm{x}^{\text{src}}_t|\mathrm{x}^{\text{src}}, \mathrm{y}^{\text{src}})$, where the technique is also employed in the controllable generation described in Section I of [C1]. Note that, since we employ the deterministic DDIM process, drawing multiple samples does not change anything.
>
> Q2. Classifier free guidance (CFG) v.s. CSG w/o mixup
>
> We test CFG using five different values of the guidance scale $s$, {3,4,5,6,7}, to compare them with CSG w/o mixup. Due to the space constraints, we only report the best result of CFG for each task in Table A4, where the results of CSG w/o mixup and DDIM or CFG (s=5) are presented in Table 2. The tables imply that CSG w/o mixup always outperforms CFG with the best hyperparameter $s$ ($s$ = 3 for all tasks). In case of the dog-to-dog with glasses task, we do not report the BG-LPIPS score, since the background region can be easily preserved for the task. We appreciate you for a good suggestion, and will add all experimental results of $s$ from 3 to 7 in the final version.
>
> Table A4: Quantitative results of CFG with the best classifier guidance scale $s$ from the LAION 5B dataset for various tasks using the pre-trained Stable Diffusion model.
>
> | | CS (↑)  | SD (↓) | BG-LPIPS (↓)  | RD (↓) |
> |:----------------:|:---------------:|:---------------:|:---------------:|:---------------:|
> |    Cat → Dog  | 0.2938  | 0.0582 | 0.3442 |0.2480 |
> |   Dog → Cat  |  0.2894 | 0.0611 | 0.3373 | 0.2991 |
> |    Wolf → Lion  | 0.2991  | 0.0611 | 0.3728 | 0.5495 |
> |    Zebra → Horse  | 0.2930  | 0.0788 | 0.3923  | 0.8353 |
> |    Dog → Dog w/ glasses  |  0.3124 | 0.0497  | - | 0.2453 |
>
> Q3. DDIM w/ mixup v.s. DDIM
>
> We report the results of DDIM combined with our mixup strategy, denoted by DDIM w/ mixup, in Table A5 to show the effectiveness of cross-attention mixup, where the results of DDIM are presented in Table 2. The tables demonstrate that DDIM w/ mixup always outperforms DDIM except the only one case in the zebra-to-horse task. Moreover, Table 2 indicates that our mixup is also effective when combined with the proposed conditional score guidance.
>
> Table A5: Quantitative results of DDIM combined with the proposed mixup strategy from the LAION 5B dataset for various tasks using the pre-trained Stable Diffusion model. The black bold-faced number represents better performance compared with the results of DDIM presented in Table 2.
>
> |  | CS (↑)  | SD (↓) |  BG-LPIPS (↓)  | RD (↓) |
> |:----------------:|:---------------:|:---------------:|:---------------:|:---------------:|
> |  Cat → Dog  |    **0.2923**   | **0.0697** | **0.3746** | **0.4310** |
> |  Dog → Cat    |    **0.2914**   | **0.0717** | **0.3538** | **0.4261** |
> |  Wolf → Lion   |    **0.2995**   | **0.0664** | **0.3875** | **0.7821** |
> |  Zebra → Horse    |   0.2986   | **0.0885** | **0.4074** | **0.8563** |
> |  Dog → Dog w/ glasses   |    **0.3196**   | **0.0572** | - | **0.3208** |
>
> Q4. Visualization of the covariance
>
> Figure E in the rebuttal document (Rebuttal_CSG.pdf) visualizes the precision matrix in Eq. (14).
> Note that the two values of $\mathrm{x}^{\text{src}}_t$ and $\mathrm{x}^{\text{tgt}}_t$ at the object region become more different when the reverse timestep $t$ is closer to 0. This implies that our estimation $\mathrm{x}^{\text{tgt}}_t$ for the true mean estimation of $p(\mathrm{x}^{\text{src}}_t|\mathrm{x}^{\text{tgt}}_t, \mathrm{y}^{\text{tgt}})$ can be imprecise at the object region when the timestep is close to 0.
> However, we can ignore the error since the corresponding precision values are set to 0 as visualized in the figure.
> Also, considering Eq. (16), the role of the precision (or the inverse of the covariance) adaptively encourages $\mathrm{x}^{\text{tgt}}_t$ to become close to $\mathrm{x}^{\text{src}}_t$ depending on the precision values at the regions, which is a well-suited formulation for image-to-image translation tasks. We appreciate you for a good suggestion, and we will add the visualization in the final version.
>
> Q5. Comparison between CSG w/o mixup and existing guidance methods
>
> To compare CSG w/o mixup with existing guidance methods, please refer to the result of Pix2Pix-Zero and Prompt-to-Prompt presented in Table 1 and the results of CSG w/o mixup in Table 2. As presented in the tables, CSG w/o mixup outperforms the two guidance methods in most cases. Moreover, Figure C in the rebuttal document (Rebuttal_CSG.pdf), which contains compatible results with Figure 4 of the main paper, demonstrates that CSG w/o mixup achieves better qualitative results.
>
> Q6. Presentation
>
> We will carefully revise the manuscript to reflect your comments for adding a bridging paragraph between line 35 and line 36, and correcting the typo in Algorithm 1.
>
> Q7. Dimension of the covariance
>
> We replicate the same value of Ω for C times to match the dimension.
>
> Q8. Analysis and detailed description of relational distance
>
> For the detailed description of RD, we have already presented it in Section A.3 of the supplementary material. Different from RD, other metrics compare at an individual instance level. The instance level comparisons may be insufficient to evaluate existing algorithms, and it is important to consider the entire structures given by the two sets by measuring the relational set information as RD. Therefore, we believe that RD can provide a more comprehensive assessment of the performance.
>
> Q9. Additional sampling cost of the conditional guidance
>
> Due to the space constraints, please refer to our response to Q2 of Reviewer JnRy.
>
> Reference
>
> [C1] Y. Song et al., Score-Based Generative Modeling through Stochastic Differential Equations, ICLR 2021.

---

> > ### Author Response · Authors · 2023-08-14
> > **After Rebuttal**
> >
> > Dear Reviewer nbbt,
> >
> > Because the end of discussion period is approaching, we kindly ask you whether our response is helpful to clarify you or not. Also, if you have any questions or additional comments, please do not hesitate to contact us. We thank you for your time and efforts to review our paper.
> >
> > Best wishes,
> >
> > Authors

---

> > > ### Comment · Reviewer_nbbt · 2023-08-21
> > >
> > > I would keep my initial rating since the Author response resolved my major concerns.
> > >
> > > The biggest concerns I had were
> > > 1. Validity of the derivation
> > > 2. Missing experiments
> > >
> > > which are answered well.
> > >
> > > Thank you for the rebuttal.

---

> > > > ### Author Response · Authors · 2023-08-21
> > > > **Thanks for the comment**
> > > >
> > > > We thank you and we will revise the main paper to reflect all comments including the missing experiments.
> > > >
> > > > If you have any questions or additional comments, please do not hesitate to contact us.
> > > >
> > > > Best wishes,
> > > >
> > > > Authors

---

### Official Review · Reviewer_fXLA · 2023-07-08

**Soundness:** 3 good
**Presentation:** 3 good
**Contribution:** 3 good
**Rating:** 5
**Confidence:** 4

**Summary:**

This paper proposes a new score function for text driven image to image translation. The core idea is to estimate score function conditioned on both original image and the target prompt. The score function can be decomposed into two parts, one is from the target prompt and the other one is guiding term for target image generation. In addition, they also use a trick to get better masks for better preserving non-interested regions such as background.

**Strengths:**

The idea is sound both theoretically and empirically. Their writing is clear.

**Weaknesses:**

1, why do they choose those 5 tasks to evaluate? I know they are from prior work, but I am a bit concern that they can not demonstrate the generalizability of the method.

2, I hope they could show random samples so that we can see the average performance of their method.



**Questions:**

na

---

> ### Author Rebuttal · Authors · 2023-08-10
>
> We truly thank you for your constructive and positive comments and below are our responses to the main questions.
>
> Q1. Concern about the generalizability of CSG
>
> In the main paper, our paper focused on testing CSG on local editing tasks such as cat-to-dog, dog-to-cat, wolf-to-lion, zebra-to-horse, and dog-to-dog with glasses.
> In addition to the local editing tasks, we employ global editing tasks, street-to-snowy street and drawing-to-realistic photo tasks, to compare CSG with Pix2Pix-Zero. As presented in Table A2 and Table A3, the proposed method outperforms Pix2Pix-Zero in terms of SD, LPIPS, and RD even with faster translation although Pix2Pix-Zero achieves slightly higher values of CS.
> In addition, we visualize generated samples in Figure A-1 and Figure A-2 in the rebuttal document (Rebuttal_CSG.pdf), which demonstrate that the proposed method also achieves better performance on the tasks.
> For the global editing tasks, note that we replace BG-LPIPS with LPIPS, where LPIPS [B1] measures the perceptual similarity using the entire source and target images, which is more suitable for the global editing tasks.
> We appreciate you for a good suggestion, and we will add the results in the final version.
>
>
> Table A2: Quantitative results to compare with Pix2Pix-Zero using the pre-trained Stable Diffusion and its synthetic images for Street $\rightarrow$ Snowy Street task. The black bold-faced number represents the best performance in each column.
>
>
> | | CS ($\uparrow$)  | SD ($\downarrow$) | LPIPS ($\downarrow$)  | RD ($\downarrow$) |
> |:----------------:|:---------------:|:---------------:|:---------------:|:---------------:|
> |    Pix2Pix-Zero  |  **0.3215**  | 0.0186 | 0.2345 | 0.1436 |
> |   CSG    |    0.3125   | **0.0166** | **0.2077** | **0.1340** |
>
>
>
> Table A3: Quantitative results to compare with Pix2Pix-Zero using the pre-trained Stable Diffusion and its synthetic images for Drawing $\rightarrow$ Realistic photo task.
>
> | | CS ($\uparrow$)  | SD ($\downarrow$) | LPIPS ($\downarrow$)  | RD ($\downarrow$) |
> |:----------------:|:---------------:|:---------------:|:---------------:|:---------------:|
> |    Pix2Pix-Zero  |  **0.2997**  | 0.0414 | 0.2783 | 0.3933 |
> |   CSG    |    0.2966   | **0.0263** | **0.0722** | **0.2190** |
>
>
> Q2. Qualitative samples
>
> We present additional samples in the rebuttal document (Rebuttal_CSG.pdf), and please refer to the figures.
>
>
> Reference
>
> [B1] Z. Richard et al., The unreasonable effectiveness of deep features as a perceptual metric, CVPR 2018.

---

> > ### Author Response · Authors · 2023-08-14
> > **After Rebuttal**
> >
> > Dear Reviewer fXLA,
> >
> > Because the end of discussion period is approaching, we kindly ask you whether our response is helpful to clarify you or not.
> > Also, if you have any questions or additional comments, please do not hesitate to contact us.
> > We thank you for your time and efforts to review our paper.
> >
> > Best wishes,
> >
> > Authors

---

### Official Review · Reviewer_JnRy · 2023-07-14

**Soundness:** 3 good
**Presentation:** 2 fair
**Contribution:** 3 good
**Rating:** 5
**Confidence:** 5

**Summary:**

In this paper, the authors propose a new method that can perform image-to-image translation through a pretrained text-to-image. They propose a new cross-attention map mixing technique and a new conditional score guidance function to tackle this problem. The method introduced in this paper does not require additional training. The authors have also shown strong qualitative results demonstrating various application scenarios.

**Strengths:**

The proposed method does not require model architecture modifications or any training of the pretrained model. The qualitative results also look very promising. The paper is very coherently written.

**Weaknesses:**

1. Equation 3 is the same as the paper “Xuan Su, Jiaming Song, Chenlin Meng, Stefano Ermon. Dual Diffusion Implicit Bridges for Image-to-Image Translation. ICLR 2023”, which the authors did not cite or compare.
2. There are a lot of approximations and conjectures in the theory which are not explained anywhere in the main paper or the appendix.
3. The presentation of the tables is a little bit misleading. It almost seems like authors choose to denote both the best and the second best results in each category just so that their CS scores won’t look too bad.
4. The authors invented a new metric to evaluate the performance. However, there is no detail of this new metric mentioned in the main paper and readers will have to refer to the appendix to understand what this metric is exactly.
5. There is no discussion of the limitations of the method, nor of any potential negative societal impact.
6. The ablation study is very limited and it doesn’t include experiments to study the effects of their newly proposed conditional score guidance.

**Questions:**

Related to Weakness (1), can the authors compare their method to “ual Diffusion Implicit Bridges for Image-to-Image Translation”?
How long does it take to generate one image?

**Limitations:**

There is no discussion of the limitations of the method, nor of any potential negative societal impact.

---

> ### Author Rebuttal · Authors · 2023-08-10
>
> We truly thank you for your constructive comments and below are our responses to the main questions.
>
>
> Q1. Citation and comparison with DDIB [A1]
>
> We omitted the reference for Eq. (3) accidentally since it is a basic equation introduced in the DDIM paper, which was also used in [A1]. We are sorry for the missing citation but we did not claim that the derivation or usage of the equation is our contribution. Note that the naive method in Section 3.1 of the main paper refers to [A1] and used DDIB as our baseline.
> Also, we have already compared our method with DDIB, which is actually referred to as DDIM in both the main paper and supplementary material. We thank you for the comment and will clarify the reference issue in the final version.
>
>
>
>
> Q2. Inference time compared with DDIB or DDIM
>
> In order to observe the realistic speed of each algorithm, we measure the wall clock time using a NVIDIA A100 GPU with a single image.
> Although the naive DDIM translation algorithm or DDIB retains the fastest inference time as presented in Table A1, the framework achieves poor generation results as mentioned in the main paper and supplementary material.
> For the theoretical inference comparison, note that CSG approximately requires an extra 0.5x inference cost of DDIB since CSG needs an extra computation of reversing the latent using the source prompt embedding different from DDIB.
> However, in case of CSG and Pix2Pix-Zero, there is a disparity between theoretical and practical inference costs since the communication cost for copying cross-attention maps from the GPU memory to the CPU memory is not negligible.
> In CSG, in order to save the GPU memory, our algorithm applies the resizing operation in CPU to the cross-attention maps for computing the smooth content mask, which further makes the inference speed become slower.
> Therefore, the practical speed can be reduced if we have enough GPU memory.
>
>
> Table A1: Computational cost of the proposed method compared to DDIB and Pix2Pix-Zero.
>
> | | DDIB  | Pix2Pix-Zero  | CSG w/o mixup | CSG |
> |:----------------:|:---------------:|:---------------:|:---------------:|:---------------:|
> |    time/image (s)  | 5.129  | 28.647 | 19.791 | 25.736 |
>
>
>
> Q3. A lot of approximations and conjectures in the theory
>
> We used two approximations for CSG, one of which is about the score function approximation mentioned in Eq. (12) of the main paper.
> Note that all previous methods relying on text-to-image diffusion models including the simple translation algorithm employ the same approximation.
> Also, we employ the only one additional approximation in Eq. (8) of the main paper, using the deterministic DDIM sample $\hat{\mathrm{x}}^{\text{src}}_t$ drawn from $p(\mathrm{x}^{\text{src}}_t|\mathrm{x}^{\text{src}}, \mathrm{y}^{\text{src}})$, which is also described in line 164 of the main paper.
> Thanks to the additional approximation for the proposed guidance, CSG outperforms the previous methods.
>
>
>
>
>
>
> Q4. Misleading presentation of Tables
>
> We simply highlight the best and second-best performance in each metric, and kindly ask you whether our response is helpful to clarify you or not.
>
>
>
>
>
>
> Q5. Details of relational distance
>
> We did not mention in the main paper about providing the detailed information regarding RD in the supplementary material although the detail is described in the supplementary material.
> We will carefully revise the manuscript to reflect your comment.
>
>
> Q6. Discussion of limitations and potential negative societal impact
>
> Our method can fail to edit images with complex prompts due to the incompetence of pre-trained text-to-image diffusion models. As other text-driven image-to-image translation methods, the proposed method also has another limitation that it can not be applied to complex tasks such as enlarging the object or moving the object, where it would be an interesting work to solve the difficult tasks. For the potential negative social impact, our method can generate harmful or misleading contents due to the pre-trained model. We will add the limitations and negative social impacts in the final version.
>
>
>
> Q7. Analysis on the newly proposed conditional score guidance
>
> We already provided the ablation study results in Table 2 of the main paper.
> The table implies that the conditional score guidance without using the proposed mixup denoted by CSG w/o Mixup is helpful to preserve structures of source images compared with the naive translation algorithm.
> Also, we present the qualitative results in Figure C in the rebuttal document (Rebuttal_CSG.pdf), which contains compatible results with Figure 4 of the main paper.
> Considering the two figures, our conditional score guidance significantly outperforms DDIM.
>
>
> Reference
>
> [A1] X. Su et al., Dual Diffusion Implicit Bridges for Image-to-Image Translation, ICLR 2023.

---

> > ### Author Response · Authors · 2023-08-14
> > **After Rebuttal**
> >
> > Dear Reviewer JnRy,
> >
> > Because the end of discussion period is approaching, we kindly ask you whether our response is helpful to clarify you or not.
> > Also, if you have any questions or additional comments, please do not hesitate to contact us.
> > We thank you for your time and efforts to review our paper.
> >
> > Best wishes,
> >
> > Authors

---

> > ### Comment · Reviewer_JnRy · 2023-08-19
> > **Thank you for your response**
> >
> > Thank you for your clarification. My major concerns have been addressed in the rebuttal. I would like to change my rating from 3 to 5.

---

> > > ### Author Response · Authors · 2023-08-20
> > > **Thanks for the comment**
> > >
> > > We appreciate you, and we will revise the main paper to reflect your comments.
> > >
> > > If you have any questions or additional comments, please do not hesitate to contact us.
> > >
> > >
> > > Best wishes,
> > >
> > > Authors

---

### Author Rebuttal · Authors · 2023-08-10

We sincerely thank all reviewers for their time and efforts in reviewing our main paper.
We have attached our visualization results in the document "Rebuttal_CSG.pdf", and please refer to the qualitative results.

---

### Comment · Area_Chair_4UpK · 2023-08-17
**Kindly request to respond to rebuttal from authors**

Dear reviewers of #5715,

Thanks for your time on reviewing this work. Since authors have provided detailed feedback regarding the proposed concerns, it would be great for you to check the response and see if your concern is addressed. Given that the deadline of discussion between authors and reviewers is approaching in a few days, your timely feedback will allow authors to further clarify.

Thanks Reviewer rP2j who already responded.

Yours,
AC

---

### Author Response · Authors · 2023-08-19
**Summary of Author Responses**

We sincerely thank all reviewers for their constructive and positive comments and we present the summary of our responses to each reviewer as below.

Q1. Concern about the generalizability of CSG

In the main paper, our paper focused on testing CSG on local editing tasks such as cat-to-dog, dog-to-cat, wolf-to-lion, zebra-to-horse, and dog-to-dog with glasses.
In addition to the local editing tasks, we employ global editing tasks, street-to-snowy street and drawing-to-realistic photo tasks, to compare CSG with Pix2Pix-Zero. As presented in Table A2 and Table A3, the proposed method outperforms Pix2Pix-Zero in terms of SD, LPIPS, and RD even with faster translation although Pix2Pix-Zero achieves slightly higher values of CS.
In addition, Figure A-1 and Figure A-2 demonstrate that the proposed method also achieves better performance on the tasks.


Q2. Comparison with Plug-and-Play

Table 1 and Table A7  imply that the proposed method outperforms Plug-and-Play in most cases. Also, Figure D in the rebuttal document and Figure 1 imply that CSG achieves much better qualitative results. Furthermore, considering the qualitative results in Figure C and D, the proposed method even without using the cross-attention mixup outperforms Plug-and-Play.




Q3. Limited editing capabilities

We tried to follow the experiment protocol of prior works [Prompt-to-Prompt, Pix2Pix-Zero] by evaluating CSG on object-centric editing tasks. In addition to the object-centric editing tasks, our method can be extended to global editing tasks. To validate the property, we test CSG and Pix2Pix-Zero using global editing tasks, street-to-snowy street and drawing-to-realistic photo tasks. As presented in Table A6, CSG outperforms Pix2Pix-Zero in terms of SD, LPIPS, and RD even with faster translation although Pix2Pix-Zero achieves slightly higher values of CS. In addition to the quantitative results, Figure A-1 and A-2 demonstrate that the proposed method achieves better performance on the tasks.


Q4. Incremental novelty

Our algorithm is somewhat related to Pix2Pix-Zero in the sense that CSG and Pix2Pix-Zero employ the cross-attention layers in the noise prediction network for text-driven image-to-image translation tasks. However, the two methods are completely different, where CSG uses mixup strategy in cross-attention layers while Pix2Pix-Zero takes a gradient step to reduce the distance between the cross-attention maps given by the reverse process of $\mathrm{x}^{\text{src}}_t$ and $\mathrm{x}^{\text{tgt}}_t$, which requires an additional backpropagation step and leads to slower inference. Moreover, as mentioned by Reviewer fXLA, Reviewer nbbt, and Reviewer rP2j, we propose a principled technique for text-driven image translation based on the conditional score function with reasonable motivations. Such ideas have not been addressed before and are sound both theoretically and empirically, so we believe that our paper has sufficient novelty and contribution.


Q5. Additional comparison with Pix2Pix-Zero

Different from CSG, it is hard to apply Pix2Pix-Zero for dissimilar object-centric tasks such as hose-to-eiffel tower as presented in the right examples of Figure B since Pix2Pix-Zero enforces the shape matching through the cross-attention layers. Also, Figure A-1, A-2, and B demonstrate that CSG outperforms Pix2Pix-Zero.


Q6. (Ablation) Classifier free guidance (CFG) v.s. CSG w/o mixup

We test CFG using five different values of the guidance scale $s$, {3,4,5,6,7}, to compare them with CSG w/o mixup. Table A4 and Table 2imply that CSG w/o mixup always outperforms CFG with the best hyperparameter $s$ ($s$ = 3 for all tasks).


Q7. (Ablation) DDIM w/ mixup v.s. DDIM

To show the effectiveness of cross-attention mixup, we report the results of DDIM combined with our mixup strategy, denoted by DDIM w/ mixup, in Table A5, where the results of DDIM are presented in Table 2. The tables demonstrate that DDIM w/ mixup always outperforms DDIM except the only one case in the zebra-to-horse task. Moreover, Table 2 indicates that our mixup is also effective when combined with the proposed conditional score guidance.

---

> ### Author Response · Authors · 2023-08-19
> **Summary of Author Responses (2)**
>
> Q8. Computational cost in terms of time and memory
>
> In order to observe the realistic speed of each algorithm, we measure the wall clock time using a NVIDIA A100 GPU with a single image while checking the memory consumption.
> Although the naive DDIM translation algorithm retains the fastest inference time as presented in Table A8, it achieves poor generation results as mentioned in the main paper and supplementary material.
> For the theoretical inference comparison, note that CSG approximately requires an extra 0.5x inference cost of DDIM since CSG needs an extra computation of reversing the latent using the source prompt embedding different from DDIM.
> However, in case of CSG and Pix2Pix-Zero, there is a disparity between theoretical and practical inference costs since the communication cost for copying cross-attention maps from the GPU memory to the CPU memory is not negligible.
> In CSG, in order to save the GPU memory, our algorithm applies the resizing operation in CPU of the cross-attention maps for computing the smooth content mask, which further makes the inference speed become slower.
> Therefore, the practical speed can be reduced if we have enough GPU memory.
>
>
> Q9. Visualization of the covariance
>
> Figure E visualizes the precision matrix in Eq. (14) and note that the two values of $\mathrm{x}^{\text{src}}_t$ and $\mathrm{x}^{\text{tgt}}_t$ at the object region become more different when the reverse timestep $t$ is closer to 0. This implies that our estimation $\mathrm{x}^{\text{tgt}}_t$ for the true mean estimation of $p(\mathrm{x}^{\text{src}}_t|\mathrm{x}^{\text{tgt}}_t, \mathrm{y}^{\text{tgt}})$ can be imprecise at the object region when the timestep is close to 0.
> However, we can ignore the error since the corresponding precision values are set to 0 as visualized in the figure.
> Also, considering Eq. (16), the role of the precision (or the inverse of the covariance) adaptively encourages $\mathrm{x}^{\text{tgt}}_t$ to become close to $\mathrm{x}^{\text{src}}_t$ depending on the precision values at the regions, which is a well-suited formulation for image-to-image translation tasks.
>
>
> Q10. Other pre-trained text-to-image diffusion models
>
> Our framework generalizes well to the diffusion models. To show that the proposed method generalizes well, we visualize generated images given by CSG using LDM in Figure G.
> The figure demonstrates that our conditional guidance method can generalize well.
> Note that LDM is similar to Stable Diffusion, however the training data of LDM and its resolution are different from that of Stable Diffusion.
> In case of other pretrained text-to-image diffusion models such as DALL-E-2 and Imagen, they are not publicly available, so we can not test CSG using the pre-trained text-to-image diffusion models.
>
>
> Q11. Discussion of limitations and potential negative societal impact
>
> Our method can fail to edit images with complex prompts due to the incompetence of pre-trained text-to-image diffusion models. As other text-driven image-to-image translation methods, the proposed method also has another limitation that it can not be applied to complex tasks such as enlarging the object part task or moving the object, where it would be an interesting work to solve the difficult tasks. For the potential negative social impact, our method can generate harmful or misleading contents due to the pre-trained model.
>
> We will carefully revise our paper to reflect your comments.

---

### Decision · Program_Chairs · 2023-09-21

**Decision:**

Accept (poster)

**Comment:**

This work focuses on the image-to-image task using diffusion models and introduced two novel components, i.e., conditional score guidance and cross-attention mixup. Reviewers are unanimously positive (though not strong) by acknowledging the contribution and its effectiveness, without obvious objections. The motivation behind CSG takes both source prompt and source image into account, which is significantly different from previous works that only considers the target prompt and of good value to inspire the image editing research field. Extensive experiments verify the advantage of proposed methods. In addition, a new metric for evaluation is introduced and demonstrated well. Therefore, after checking all comments and discussions, the area chair made a decision of acceptance. Some concerns including adding limitations and necessary experimental results presented in the rebuttal should be addressed in the revised version.